# Sulfur oxidation and reduction are coupled to nitrogen fixation in the roots of the salt marsh foundation plant *Spartina alterniflora*

J. L. Rolando[1], M. Kolton [1,2], T. Song [1], Y. Liu[1,3], P. Pinamang[1], R. Conrad[1], J. T. Morris[4], K. T. Konstantinidis [1,5] & J. E. Kostka [1,6,7] ✉

Heterotrophic activity, primarily driven by sulfate-reducing prokaryotes, has traditionally been linked to nitrogen fixation in the root zone of coastal marine plants, leaving the role of chemolithoautotrophy in this process unexplored. Here, we show that sulfur oxidation coupled to nitrogen fixation is a previously overlooked process providing nitrogen to coastal marine macrophytes. In this study, we recovered 239 metagenome-assembled genomes from a salt marsh dominated by the foundation plant *Spartina alterniflora*, including diazotrophic sulfate-reducing and sulfur-oxidizing bacteria. Abundant sulfur-oxidizing bacteria encode and highly express genes for carbon fixation (*RuBisCO*), nitrogen fixation (*nifHDK*) and sulfur oxidation (oxidative-*dsrAB*), especially in roots stressed by sulfidic and reduced sediment conditions. Stressed roots exhibited the highest rates of nitrogen fixation and expression level of sulfur oxidation and sulfate reduction genes. Close relatives of marine symbionts from the *Candidatus* Thiodiazotropha genus contributed ~30% and ~20% of all sulfur-oxidizing *dsrA* and nitrogen-fixing *nifK* transcripts in stressed roots, respectively. Based on these findings, we propose that the symbiosis between *S. alterniflora* and sulfur-oxidizing bacteria is key to ecosystem functioning of coastal salt marshes.

The microbial communities closely associated with plant hosts (i.e., plant microbiota) have received substantial attention due to their role in plant nutrient acquisition, phytohormone synthesis, prevention of soil-borne disease, and detoxification of the rhizosphere and root environment[1,2]. Most plant microbiome studies have been performed in terrestrial ecosystems with an emphasis on agricultural plants[2]. Plant microbiota from vegetated coastal ecosystems (i.e., seagrass meadows, mangroves, and salt marshes) remain understudied, even though they play a key role in global climate regulation and the cycling of major nutrients[3,4]. Despite their high ecological value, little is known

about how plant-microbe interactions contribute to the functioning of coastal marine ecosystems, their resilience to climate change, and the provisioning of ecosystem services.

Previous studies have shown that the root zone of coastal marine plants is a hotspot for the cycling of carbon, nitrogen, and sulfur[5–10]. Furthermore, sulfate-reducing and sulfur-oxidizing bacteria have been shown to be overrepresented in the core root and rhizosphere microbiome of seagrass and salt marsh plants, including that of the foundation salt marsh plant *Spartina alterniflora*[11,12]. Sulfate reduction, an anaerobic respiration pathway, represents a dominant terminal

[1]Georgia Institute of Technology, School of Biological Sciences, Atlanta, GA 30332, USA. [2]French Associates Institute for Agriculture and Biotechnology of Drylands, Ben-Gurion University of the Negev, Beer Sheva, Israel. [3]The Pennsylvania State University, Department of Civil & Environmental Engineering, University Park, PA 16802, USA. [4]Belle Baruch Institute for Marine & Coastal Sciences, University of South Carolina, Columbia, SC 29201, USA. [5]Georgia Institute of Technology, School of Civil and Environmental Engineering, Atlanta, GA 30332, USA. [6]Georgia Institute of Technology, School of Earth and Atmospheric Sciences, Atlanta, GA 30332, USA. [7]Center for Microbial Dynamics and Infection, Georgia Institute of Technology, Atlanta, GA 30332, USA. ✉ e-mail: joel.kostka@biology.gatech.edu

electron-accepting process coupled to the breakdown of organic matter in marine ecosystems[13,14]. Evidence from experimental and field research indicates that rhizospheric sulfate-reducing bacteria assimilate photosynthates from *S. alterniflora*, while fueling nitrogen fixation[5,7]. Similarly, rates of nitrogen fixation have been closely associated with organic matter degradation mainly by sulfate reduction in seagrass meadows[6,15–18]. Conversely, the oxidation of reduced sulfur compounds is a chemolithotrophic process requiring terminal electron acceptors such as oxygen, nitrate, or oxidized metals. Sulfide is a known phytotoxin, and its belowground oxidation is considered a detoxifying reaction for plants inhabiting coastal marine ecosystems[19]. Based only on amplicon sequencing of the 16S rRNA gene, sulfur oxidation coupled with nitrogen fixation was recently hypothesized as an important process for plant growth under sulfidic conditions[12]. Nitrogen is often the limiting nutrient for salt marsh plants, and stress from sulfide toxicity and anoxia further impairs root active acquisition of nitrogen[20]. Close relatives of diazotrophic sulfur-oxidizing symbionts from the *Sedimenticolaceae* family (genus: *Candidatus* Thiodiazotropha) have been shown to inhabit the roots of seagrasses and marsh plants in coastal marine ecosystems[12,21]. The *Ca.* Thiodiazotropha genus was first discovered as bacterial symbionts of lucinid clams, where they provide both fixed carbon and nitrogen to their animal host by sulfur-mediated chemolithoautotrophy[22,23]. Most studies assessing the ecology and function of sulfur chemosymbiosis have been performed within marine invertebrate hosts[22,24–27]. Cúcio et al.[28] binned genomes from four bacterial species associated with seagrass roots, including a chemolithoautotrophic sulfur-oxidizing bacteria. However, none of these genomes harbored genes for nitrogen fixation. Thus, strong evidence is lacking for the coupling of sulfur oxidation with nitrogen fixation in the roots of coastal marine plants due to the fact that the genomes, metabolic potential, and activity of diazotrophic sulfur-oxidizing bacteria have not been characterized. Limited studies have investigated the composition and abundance of sulfur-oxidizing bacteria present in the roots of coastal marine plants.

Previous studies relied on microscopy-, DNA/RNA amplicon-based community analyses or independent metagenomic and metatranscriptomic analyses[8–10,12,21,28]. Because a multi-omics approach has not yet been applied to the roots of coastal plants, genome-wide gene expression profiles of uncultured sulfur-oxidizing bacteria and the coupling of sulfur oxidation with other biogeochemical processes are still to be deciphered. In addition, although the coupling of sulfate reduction to nitrogen fixation is well established in the roots of marine plants, the metabolically-active sulfate reducers that mediate this process and their metabolic potential have not been explored with a genome-centric approach.

Ecosystem models and empirical evidence indicate that climate change is altering the hydrology, biogeochemistry, and plant community composition of coastal wetlands[29–31]. Thus, a mechanistic understanding of how environmental perturbations impact plant-microbe interactions will be critical to forecasting the resilience of coastal marine ecosystems to climate change. We hypothesize that beneficial plant-microbe interactions related to plant stress amelioration will be more prevalent in plants experiencing higher levels of salinity, reduced redox conditions, and sulfide toxicity. *S. alterniflora*-dominated salt marsh ecosystems represent an ideal natural laboratory in which to study the effects of stress on plant-microbe interactions because steep gradients in plant productivity are formed within short distances[20]. The productivity gradient is the result of decreased soil redox potential, along with increased salinity and anoxia, extending from vegetated tidal creek banks towards the interior of the marsh (Fig. 1). Higher primary productivity is reflected in tall *S. alterniflora* plants (>80 cm) growing adjacent to tidal creek banks, while smaller plants (<50 cm) inhabit the interior of the marsh. Furthermore, *S. alterniflora* physiological impairment along the stress gradient has been widely studied and evidenced by lower aboveground biomass and photosynthesis rates, decreased leaf area, and reduced energy status for root nitrogen uptake due to root anaerobic metabolism[20,32,33]. Since the root zone of the stressed short phenotype

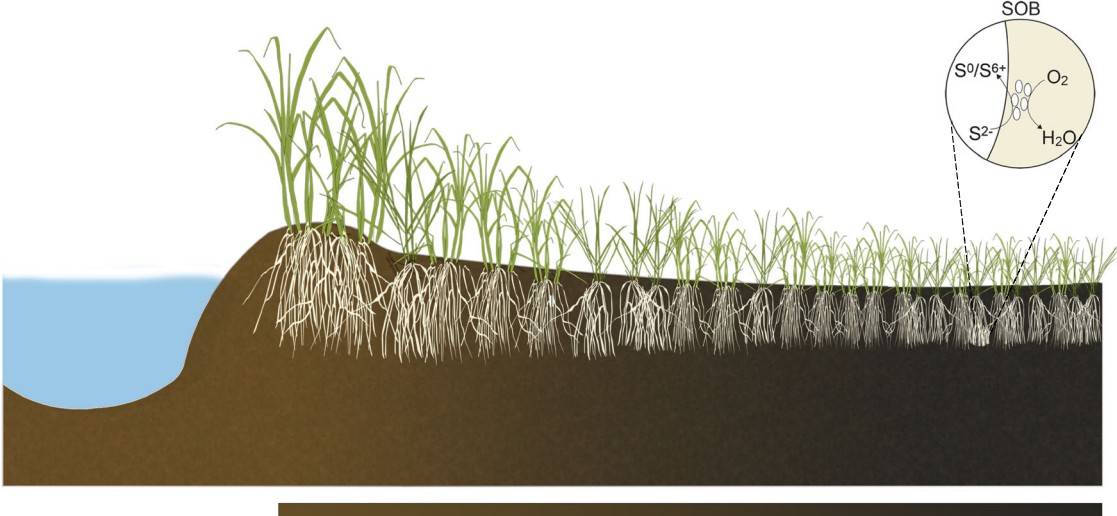

↑ Interstitial [$Fe^{2+}$]
↑ Rates of $Fe^{3+}$ reduction
↑ Bioturbation and tidal flushing
↑ Rates of coupled nitrification-denitrification
↑ Rates of SOM hydrolysis and mineralization

↑ Interstitial [$HS^-$]
↑ Porewater salinity
↑ Rates of $SO_4^{2+}$ reduction

**Fig. 1 | *Spartina alterniflora* biomass gradient as a natural laboratory.** A gradient in *S. alterniflora* aboveground biomass is commonly observed with tall plants growing at the levees next to large tidal creeks and a short phenotype of *S. alterniflora* dominating the interior of the marsh. Sediments from the tall *S. alterniflora* zone are characterized as a more oxidized environment with higher levels of iron reduction and coupled nitrification-denitrification as well as higher rates of organic matter hydrolysis and mineralization. Conversely, sediments from the short phenotype tend to be more chemically reduced, with higher rates of sulfate reduction, elevated porewater salinity, and less bioturbation and tidal flushing. Roots from the short *S. alterniflora* phenotype have been proposed to harbor sulfur-oxidizing bacteria (SOB) that benefit the plant by detoxifying the root environment.

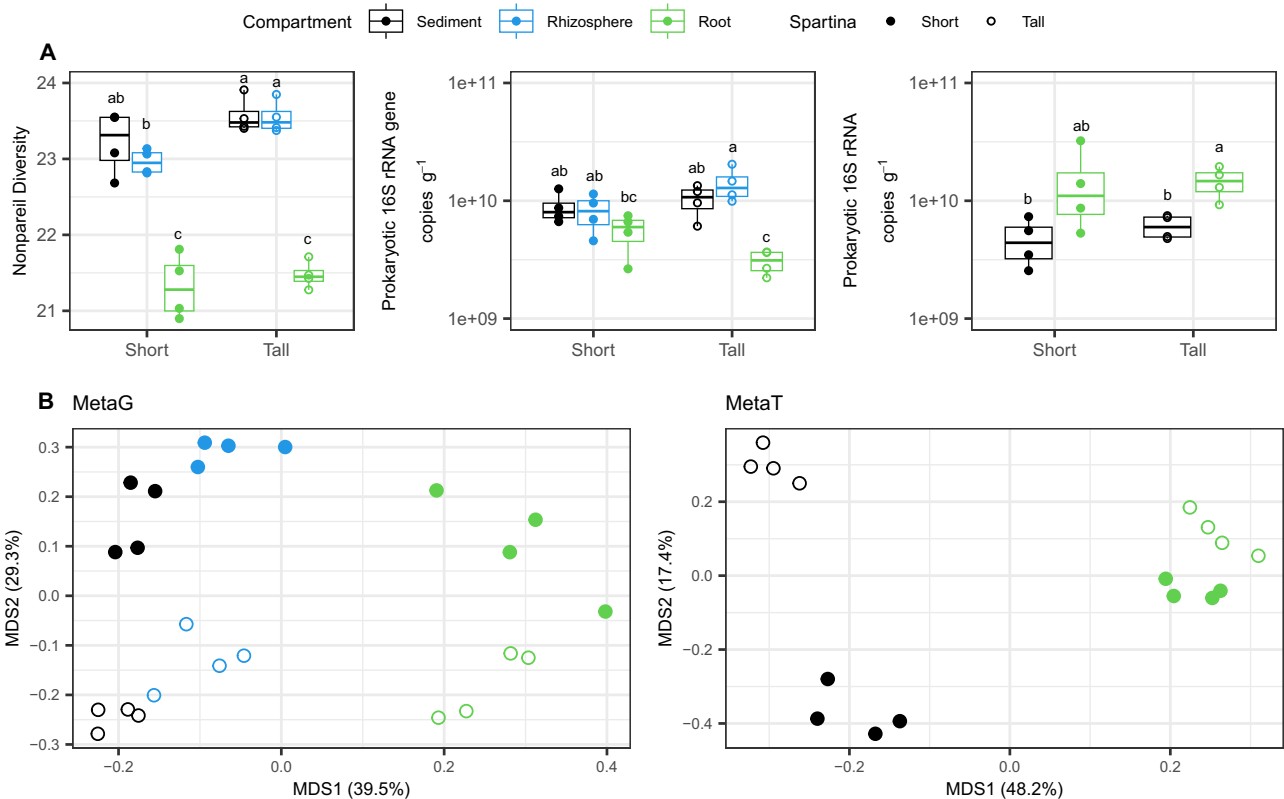

**Fig. 2 | Prokaryotic abundance, functional diversity, and activity are determined by *Spartina alterniflora* phenotype and microbiome compartment.** Metagenomic nonpareil diversity, 16S rRNA gene, and transcript abundance as quantified by qPCR and RT-qPCR, respectively (*n* = 4 per compartment and *S. alterniflora* phenotype) (**A**). Principal coordinate analysis (PCoA) ordination plot based on the Bray-Curtis dissimilatory index of functional profiles from KEGG orthology annotations (KO level) of metagenome and metatranscriptome libraries (**B**). In boxplots, boxes are defined by the upper and lower interquartile; the median is represented as a horizontal line within the boxes; whiskers extend to the most extreme data point which is no more than 1.5 times the interquartile range. Different letter indicates statistical difference based on pairwise Mann–Whitney tests (two-sided, *p*-value < 0.05).

establishes an oxic-anoxic interface conducive for sulfur oxidation, we hypothesize a stronger relationship between plant roots and diazotrophic sulfur chemosymbionts in this microhabitat of the marsh. The present study closely couples biogeochemistry with a multi-omics approach to address the following objectives: (i.), to demonstrate the coupling of sulfur oxidation with nitrogen fixation in the root environment of *S. alterniflora*, (ii.) to evaluate the effect of environmental stress on the assembly, activity, and ecological interactions of the *S. alterniflora* root microbiome, and (iii.) to infer the extent of the interaction between diazotrophic sulfur-cycling bacteria in the roots of coastal marine plants.

Here, we present metagenome-assembled genomes (MAGs) from sulfate-reducing and sulfur-oxidizing bacteria that contain and highly express genes for nitrogen fixation in the roots of *S. alterniflora*. Further, we reveal that the most dominant and active sulfur-oxidizing bacteria in the *S. alterniflora* root are closely related to lucinid clam symbionts from the *Ca.* Thiodiazotropha genus. A meta-analysis of amplicon datasets from terrestrial and coastal marine ecosystems shows that seagrass meadows, salt marshes, and mangrove plants assemble similar root microbiomes, with a high abundance of *Ca.* Thiodiazotropha species. Our results indicate that sulfur oxidation coupled to nitrogen fixation is a previously overlooked global process providing nitrogen to coastal marine ecosystems.

## Results

### Study site and environmental stress description

We studied the root microbiome of the salt marsh foundation species of the US Atlantic and Gulf of Mexico coastlines, *Spartina alterniflora*, at Sapelo Island, GA during the summers of 2018, 2019, and 2020.

A combination of prokaryotic DNA and RNA quantification, shotgun metagenomics, metatranscriptomics, and rate measurements of nitrogen fixation were performed at two contrasting environmental conditions within three compartments: sediment, rhizosphere, and root (Fig. 1). The short and tall *S. alterniflora* phenotypes represent the extremes of the plant biomass gradient formed by stress from salinity, sulfide toxicity and anoxia (Fig. 1). Four biological replicates per compartment and *S. alterniflora* phenotype were sequenced for both metagenomic and metatranscriptomic analysis. Rhizosphere samples were only used for metagenomic analysis due to limited samples for RNA extractions (Further details regarding sequencing can be found in the "Methods" section and Supplementary Data File S1).

Sediment and root microbiomes showed contrasting diversity, abundance, and functional profiles in both *S. alterniflora* phenotypes. Prokaryotic alpha diversity was greater in the sediment and rhizosphere compared to the root compartment in both *S. alterniflora* phenotypes (Fig. 2A). Similarly, quantification of the 16S rRNA gene by qPCR showed statistically lower prokaryotic abundance in root tissue compared to the sediment and rhizosphere (Fig. 2A). A stronger barrier for prokaryotic root colonization, evidenced by a steeper decrease in prokaryotic abundance in the rhizosphere-root interface, was observed in the less stressed tall *S. alterniflora* phenotype (Fig. 2A). Conversely, the root compartment of both *S. alterniflora* phenotypes presented a higher transcript copy number of the prokaryotic 16S rRNA than their sediment counterparts, indicating greater prokaryotic activity in the root zone (Fig. 2A). PERMANOVA and principal coordinate analysis of metagenomic and metatranscriptomic functional profiles indicate that environmental stress significantly affected the microbiome's potential and expressed functional repertoire (Fig. 2B,

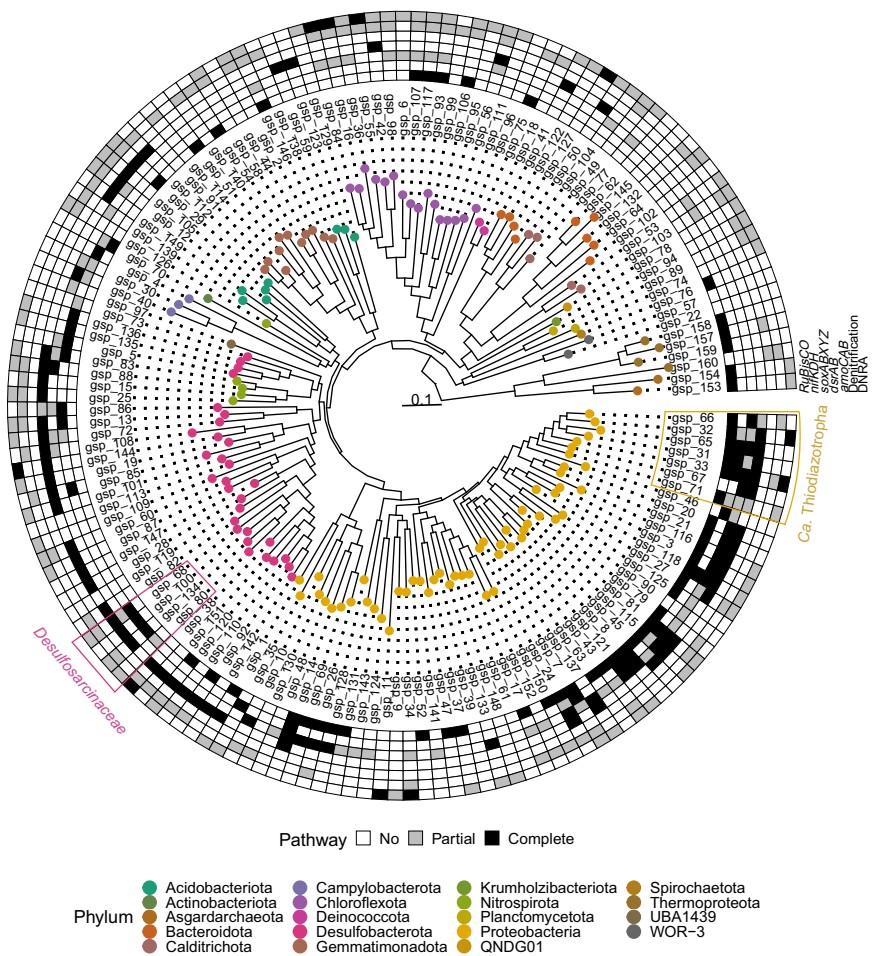

**Fig. 3 | Phylogenetic reconstruction of 160 dereplicated metagenome-assembled genomes (MAGs, >50 quality score) binned from *Spartina alterniflora* sediment, rhizosphere, and root samples.** Outer rim shows the presence/absence of genes for carbon fixation (*RuBisCO*), nitrogen fixation (*nifHDK*), thiosulfate oxidation (*soxABXYZ*), dissimilatory sulfite reduction/oxidation (*dsrAB*), nitrification (*amoCAB*), denitrification, and dissimilatory nitrate reduction to ammonium (DNRA). *Desulfosarcinaceae* and *Candidatus* Thiodiazotropha genomospecies are highlighted within pink and orange polygons, respectively.

Supplementary Table S1). Differences in potential and expressed functional metabolism were evidenced in terminal oxidases and biogeochemical pathways in the carbon, nitrogen, and sulfur cycles (Supplementary Fig. S1).

### Genome-centric multi-omics approach reveals prokaryotic sulfur metabolism is coupled to nitrogen fixation in the roots of a foundation salt marsh plant

Using a custom-designed bioinformatics pipeline, we binned 239 metagenome-assembled genomes (MAGs). Only MAGs with greater than 50 quality score (QS: Completeness – 5*Contamination) were retained. The 16S rRNA gene was binned in 68 of the 239 MAGs (Supplementary Data File S2). MAGs were dereplicated into 160 genomospecies (gsp, plural: gspp) by grouping them according to 95% average nucleotide identity (ANI) (Fig. 3). The median percentage of non-eukaryotic short reads mapping back to MAGs was 6% and 22% in the tall and short roots, respectively. High prokaryotic diversity and sequencing of eukaryotic DNA prevented the assembly and binning of a large diversity still to be described in this ecosystem. MAGs were assigned to 19 phyla. Almost half of the recovered genomes were taxonomically affiliated with the *Proteobacteria* or *Desulfobacterota* phyla (Fig. 3, further MAGs taxonomic and statistical information in Supplementary Data File S2). Taxonomic novelty was assessed using GTDB-Tk v2.1.0 with the reference database R07-RS207[34]. Recovered MAGs included 1 gspp from a previously undescribed order, 10 gspp from

undescribed families, 52 gspp from undescribed genera, 94 gspp from undescribed species, and only 3 gspp from previously described species. In order to assess the MAGs' genetic potential to perform important biogeochemical functions in the salt marsh environment, we annotated their open reading frames (ORFs) using the eggNOG database and focused on predicted genes involved in the biogeochemical cycling of carbon, nitrogen, and sulfur (Supplementary Data File S3).

A large proportion of binned *Proteobacteria*, including members of the *Ca*. Thiodiazotropha genus, contained genes for nitrogen fixation, carbon fixation through RuBisCO, as well as for dissimilatory sulfur metabolism, and thiosulfate oxidation using the soxABXYZ complex (Fig. 3). Most MAGs from the *Desulfobacterota* phylum presented genes for dissimilatory metabolism of sulfur, including members of the *Desulfosarcinaceae* family (Fig. 3). Because the dissimilatory sulfite reductase enzyme (dsrAB) can function in either sulfur reduction or oxidation, we performed a phylogenetic analysis to identify the dsrAB type encoded in our MAGs. We aligned recovered *dsrAB* genes to a reference alignment[35] and annotated the *dsrAB* type based on placement in an approximately-maximum-likelihood phylogenetic tree (Supplementary Fig. S2). All *dsrAB* genes retrieved from *Proteobacteria* gspp (*Alpha-* and *Gammaproteobacteria* gspp) were placed within the oxidative type; while *dsrAB* genes recovered from *Acidobacteriota*, *Bacteroidota*, *Chloroflexota*, *Desulfobacterota*, *Gemmatimonadota*, and *Nitrospirota* gspp clustered within the reductive bacterial *dsrAB* type (Supplementary Figs. S2, S3).

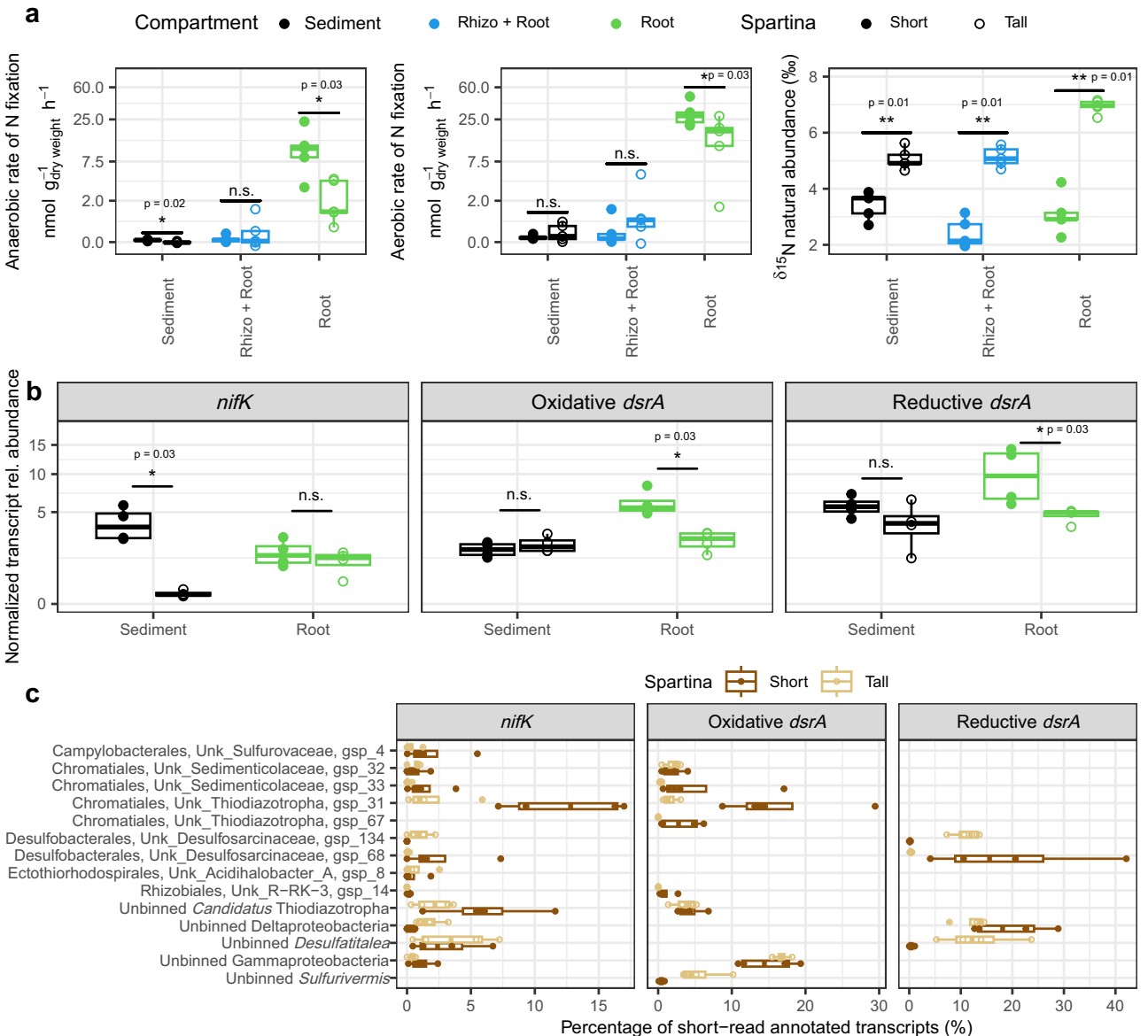

**Fig. 4 | Drivers and metagenome-assembled genomes (MAGs) associated with rates of N fixation in the salt marsh environment.** Rates of N fixation (under anoxic and oxic conditions) and $^{15}$N isotopic natural abundance per microbiome compartment and *Spartina alterniflora* phenotype ($n = 4$ per compartment and *S. alterniflora* phenotype) (**a**). $^{15}$N natural abundance is expressed as the per mille (‰) deviation from air $^{15}$N:$^{14}$N ratio ($\delta^{15}$N). Normalized transcript relative abundance of the nitrogenase gene (*nifK*), and the reductive and oxidative *dsrA* types per microbiome compartment and *S. alterniflora* phenotype (**b**). Percentage contribution of *nifK* reductive and oxidative types of *dsrA* short read transcripts mapping to the most active MAGs and unbinned scaffolds (**c**). In boxplots, boxes are defined by the upper and lower interquartile; the median is represented as a horizontal line within the boxes; whiskers extend to the most extreme data point which is no more than 1.5 times the interquartile range. Statistical significance based on non-parametric pairwise Mann–Whitney tests (two-sided). n.s. = *p*-value > 0.05, **p*-value < 0.05, ***p*-value < 0.01.

We measured rates of N fixation under oxic and anoxic conditions in sediment, a rhizosphere-root mix, and root samples using the $^{15}$N$_2$ stable isotope tracing technique. Rates were measured under oxic and anoxic conditions to capture the oxygen fluctuation experienced by microorganisms in the root zone of *S. alterniflora*. The two *S. alterniflora* phenotypes exhibited significantly higher N fixation rates in their root tissue compared to the sediment or rhizosphere-root mix under both aerobic and anaerobic conditions (Fig. 4a). Root tissue from the stressed short *S. alterniflora* phenotype showed 1.9- and 5.1-times greater N fixation rates than roots of the tall phenotype under aerobic and anaerobic conditions, respectively (Fig. 4a). In addition, natural abundance isotopic composition revealed a depletion of $\delta^{15}$N in the short phenotype compared to the tall phenotype in all assessed compartments, providing further evidence of greater nitrogen fixation in this zone of the marsh (Fig. 4a).

We used metagenomic and metatranscriptomic short reads to relate nitrogen fixation activity with multi-omics functional information. Metagenomic and metatranscriptomic reads were functionally annotated using the eggNOG database. We normalized the functional profile of each library to account for differences in sequencing effort and genome size by dividing the count matrix against the median abundance of 10 universal single-copy phylogenetic marker genes as in Salazar et al.[36]. Statistical difference between the normalized gene expression of the nitrogenase gene *nifK* and the oxidative and reductive *dsrA* types was calculated between *S. alterniflora* phenotypes across all compartments. Since gene expression within metabolic pathways was highly correlated, we used *nifK* and *dsrA* as marker genes (Supplementary Fig. S4). Even though nitrogen fixation rates were the greatest in the root compartment of the short *S. alterniflora* phenotype, no statistical significance was found in the nitrogenase gene

expression between the two phenotypes (Fig. 4b). Conversely, the root compartment of the short *S. alterniflora* phenotype had the highest normalized transcript abundance of both reductive and oxidative *dsrA* gene types (Fig. 4b). To infer which members of the *S. alterniflora* root microbiome contributed the most to nitrogen fixation, as well as to test if nitrogen fixation was coupled to sulfur metabolism, we mapped all metatranscriptomic short reads annotated as *nifK* and *dsrA* back to our MAGs as well as ORFs from assembled but not binned scaffolds. Sulfur-oxidizing *Ca*. Thiodiazotropha unbinned scaffolds and gspp (i.e., dereplicated MAGs) in the short phenotype of *S. alterniflora* contributed approximately 30% and 20% of all oxidative *dsrA* and *nifK* functionally annotated short reads, respectively (Fig. 4c). In addition, the most active sulfur-oxidizing and sulfate-reducing gspp in roots of the short phenotype (gsp 31 and gsp 68) had a positive correlation between nitrogen fixation and sulfur oxidation/reduction gene expression, respectively (Supplementary Fig. S5, Supplementary Fig. S6). In the tall phenotype, *Desulfosarcinaceae* gsp 134, and unbinned scaffolds from the *Deltaproteobacteria* class and *Desulfatitalea* genus contributed the most transcripts of the reductive *dsrA* gene (Fig. 4c). However, in the tall phenotype we were not able to map most *nifK* transcripts to MAGs, due to high diversity preventing assembly and binning. Nevertheless, using short read analysis, we found that the majority of the *nifK* transcripts in the roots of the tall phenotype were assigned to bacteria from the *Desulfobacterota* phylum, while in the short phenotype most *nifK* transcripts were affiliated with bacteria from the *Gammaproteobacteria* class or the *Desulfobacterota* phylum (Supplementary Fig. S7). Oxidative *dsrA* transcripts in the roots of the short phenotype were mostly affiliated with the *Gammaproteobacteria* class, while the reductive *dsrA* affiliated mainly with the *Desulfobacterales* and *Desulfovibrionales* orders in both phenotypes (Supplementary Figs. S8 and S9). Similarly, 16S rRNA amplicon analysis from the same sample set used for metatranscriptomic analysis showed the highest transcript abundance of gammaproteobacterial *Ca*. Thiodiazotropha in the roots of the short *S. alterniflora* phenotype when compared to the sediment of the two phenotypes and root of the tall phenotype (Supplementary Fig. S10).

## Phylogeny, genetic potential, and gene expression of microorganisms enriched in the *S. alterniflora* root microbiome

To assess the phylogenetic novelty and relation of *Sedimenticolaceae* sulfur-oxidizing MAGs from the present study in comparison to that of marine invertebrate chemosymbionts, we retrieved all publicly available *Sedimenticolaceae* genomes as reported by GTDB release R07-RS207, as well as all *Ca*. Thiodiazotropha spp. analyzed by Osvatic et al.[25]. A maximum-likelihood phylogenetic tree using 382 genes from a 400 universal marker genes database was performed in PhyloPhlAn. The phylogenetic tree used a concatenated alignment containing 14,693 amino-acid positions. We found that *S. alterniflora* root symbionts assigned to the *Ca*. Thiodiazotropha genus and an unknown genus from the *Sedimenticolaceae* family formed a monophyletic clade with lucinid clam chemosymbionts (Fig. 5). All recovered MAGs that were phylogenetically related to marine invertebrate chemosymbionts were highly abundant in the root compartment of the *S. alterniflora* short phenotype, where sulfidic conditions are found in the marsh environment (Fig. 1). To conserve phylogenetic coherence, we propose that *S. alterniflora* root symbionts assigned to the *Sedimenticolaceae* family and that formed a monophyletic group with *Ca*. Thiodiazotropha spp. are also members of this genus (Fig. 5). All *Ca*. Thiodiazotropha symbionts of *S. alterniflora* are distinct species from those previously found in lucinid clam hosts (ANI < 82% for all pairwise comparisons). Out of the 7 recovered *Ca*. Thiodiazotropha gspp, 6 had genes for carbon fixation (*RuBisCO*), 5 had the complete or partial *nifHDK* nitrogenase genes for nitrogen fixation, and all of them harbor the complete or at least partial genes for dissimilatory sulfite reductase (oxidative *dsrAB* type), and sulfur oxidation by the SOX complex

(*soxABXYZ*). All of the *Ca*. Thiodiazotropha gspp contained genes for carbon metabolism through the TCA cycle, and only two *Ca*. Thiodiazotropha gspp harbored genes for nitrate reduction (gsp 32 and gsp 33). It is possible, however, that the absence of functional genes in recovered MAGs is due to the incomplete nature of the genomes.

We calculated the gene expression profile of the most active *Ca*. Thiodiazotropha gspp (gsp 31 and gsp 33), finding that genes for sulfur oxidation through the oxidative *dsrAB*, *soxABXYZ* complex, and cytochrome c oxidase cbb3-type genes, as well as genes for carbon and nitrogen fixation (*RuBisCO* and *nifHDK*) were among the most highly transcribed (Supplementary Data Files S4 and S5). Microaerophilic oxygen reductases (i.e., cytochrome c oxidase cbb3-type and cytochrome bd ubiquinol oxidase) showed higher transcript levels compared to nitrate reductase in gsp 33 (Supplementary Data File S5).

We also found that sulfate-reducing MAGs from the *Desulfosarcinaceae* family were enriched in the root compartment of *S. alterniflora* (Supplementary Fig. S11). *Desulfosarcinaceae* gspp had contrasting preferences for root colonization, with gsp 68 mostly enriched in the roots of the short phenotype, gsp 134 preferentially enriched in the root of the tall phenotype, and gsp 80 equally abundant in both phenotypes (Supplementary Fig. S11). Of the three *Desulfosarcinaceae* gspp, all contained the complete or partial genes for nitrogen fixation, and only gsp 80 did not have the reductive *dsrAB* gene. The most transcribed genes from both gsp 68 and gsp 134 were related to dissimilatory sulfate reduction (*dsrAB* and *aprAB*, Supplementary Data Files S6 and S7).

## Plant species from coastal marine ecosystems harbor unique root microbiomes

We compiled and curated a 16S rRNA gene amplicon database from root microbiomes comprising 1182 amplicon samples from 56 plant species (complete dataset in Supplementary Data File S8). Our aim was to assess the applicability of our study's findings to global coastal vegetated ecosystems and to compare the assembly of root microbiomes in coastal marine macrophytes to that of well-studied terrestrial plants. We grouped amplicon samples into 4 broad ecosystem types: seagrass meadows, coastal wetlands (i.e., salt marshes and mangroves), freshwater wetlands, and terrestrial ecosystems. Species exchange based on the Bray-Curtis dissimilarity indices was largely explained by ecosystem type, with coastal wetland and seagrass meadow plants clustering together in an NMDS ordination (Fig. 6a). PERMANOVA analysis revealed that ecosystem type alone significantly explained 13.2% of the variation of the species exchange between analyzed samples (Supplementary Table S2). Furthermore, to explore two functional guilds that may explain the difference in microbiome assembly among ecosystem types, we assigned putative sulfate reduction and sulfur oxidation functions to our taxonomy table as performed by Rolando et al.[12]. Putative function was inferred based on homology at the genus level with prokaryotic species with known sulfur oxidation or sulfate reduction metabolism (Supplementary Data File S9). We found that the root microbiomes of both seagrass meadow and coastal wetland ecosystems were highly enriched in putative sulfate-reducing and sulfur-oxidizing bacteria (Fig. 6b, c). Furthermore, we discovered that in both seagrass meadows and coastal wetland ecosystems, amplicons showing high sequence identity to sulfur-oxidizing bacteria with the capability for nitrogen fixation (*Sedimenticolaceae* family: *Ca*. Thiodiazotropha genus) were highly abundant (Supplementary Fig. S12). In addition, amplicons showing high sequence identity to sulfate reducers from the *Desulfosarcinaceae* family, particularly those from the genus *Desulfatitalea*, were highly enriched in the roots of coastal wetland plants (Supplementary Fig. S12). The taxonomic identity of highly abundant ASVs in the root compartment of coastal marine plants match those of diazotrophic MAGs retrieved from Sapelo Island, GA, such as sulfur oxidizers of the

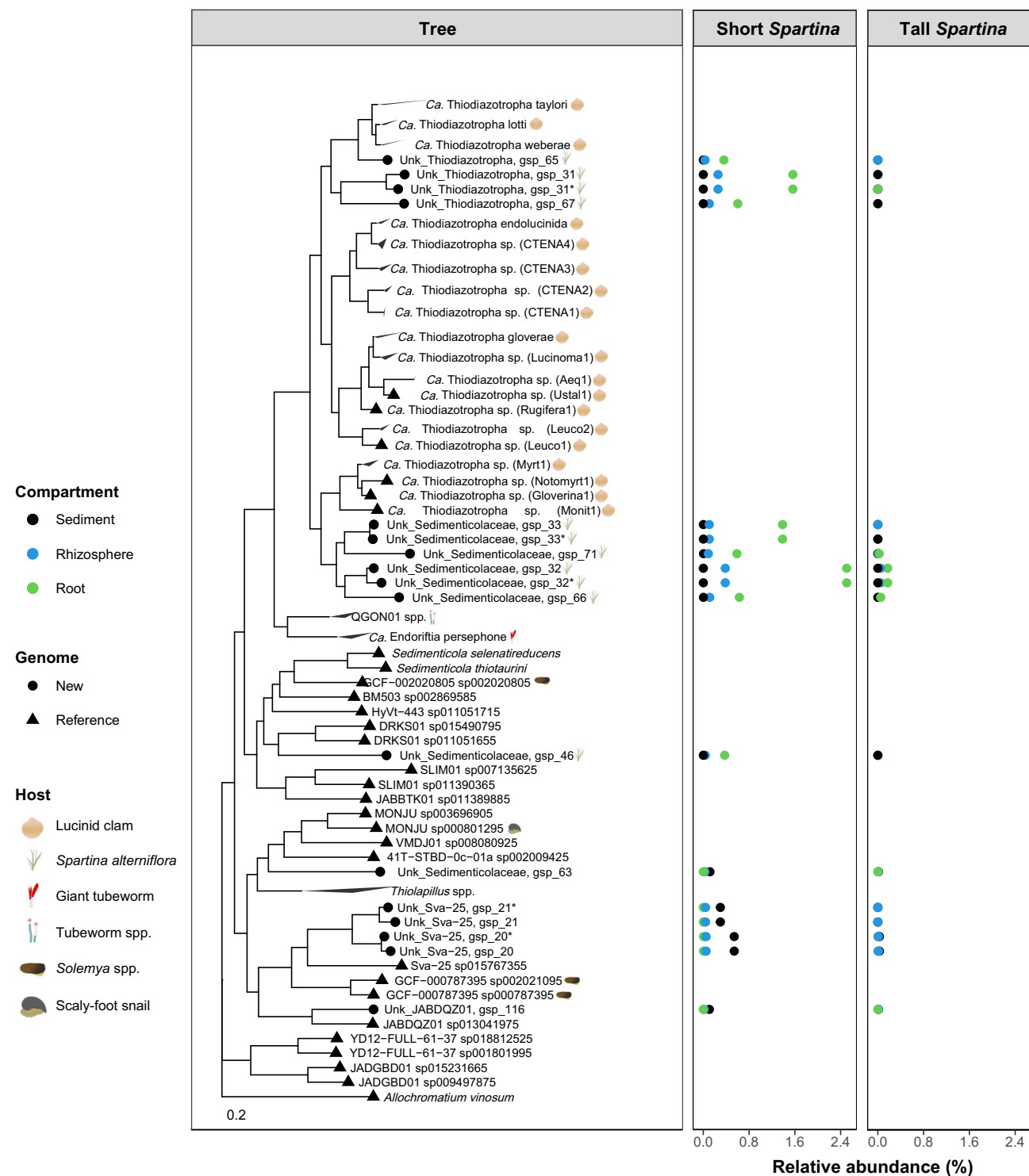

**Fig. 5 | Phylogenetic tree of the *Sedimenticolaceae* family.** Taxonomic annotation of the *Candidatus* Thiodiazotropha genus was based on Osvatic et al.[25]. The diagram next to the species name recognizes genomes recovered as symbionts of eukaryotic organisms. The average relative abundance of the metagenome-assembled genomes (MAGs) from the present study is shown in the adjacent panels per microbiome compartment and *Spartina alterniflora* phenotype. Relative abundance was calculated at the DNA level based on average coverage per position in metagenomic libraries. Purple sulfur bacteria *Allochromatium vinosum* was used as an outgroup for the phylogenetic tree.

*Sedimenticolaceae* family, and sulfate reducers of the *Desulfosarcinaceae* family.

## Discussion

Nitrogen fixation in belowground coastal vegetated ecosystems has been mainly associated with heterotrophy, and particularly with sulfate reduction[5,6,16–18]. The linkage of sulfate reduction to nitrogen fixation was proposed based on studies quantifying rates of diazotrophy with and without sulfate reduction inhibition by molybdate[6,15,16,18]. Here, we show that highly abundant sulfur-oxidizing bacteria in the roots of *S. alterniflora* also encode and highly express genes for nitrogen fixation. Furthermore, the MAG with the highest nitrogenase gene expression in

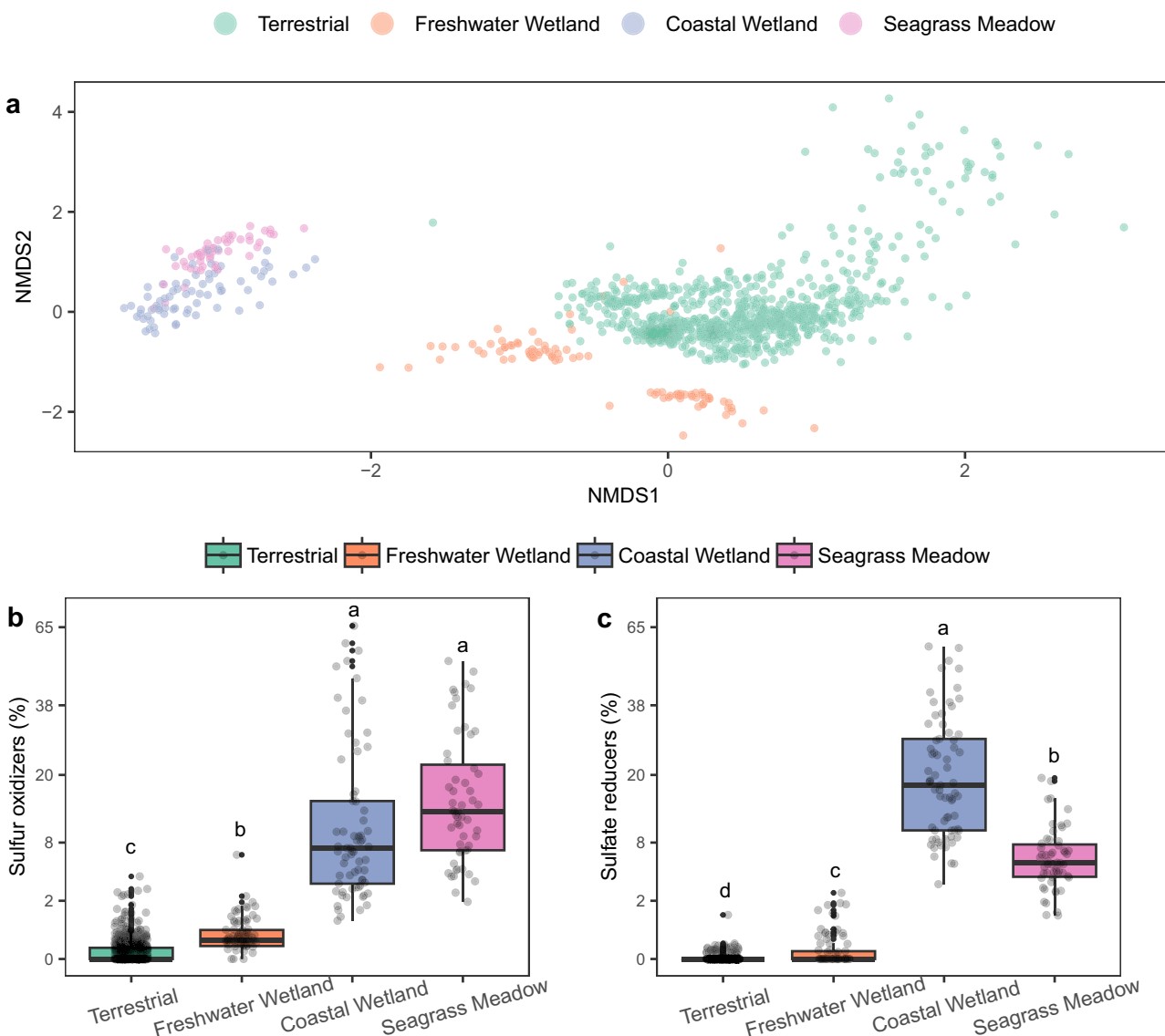

**Fig. 6 | Roots from marine-influenced ecosystems assemble a distinct microbial community enriched by bacteria known to conserve energy from sulfur metabolism.** Non-metric multidimensional scaling (NMDS) ordination plot based on the Bray-Curtis dissimilatory index of root-associated prokaryotic communities at the genus level, colored by ecosystem type (**a**). Relative abundance of putative sulfur-oxidizing (**b**) and sulfate-reducing root bacteria (**c**) by ecosystem type. Prokaryotic communities were characterized by analyzing an SSU rRNA gene amplicon dataset of 1182 samples assessing roots from 56 plant species. In box-plots, boxes are defined by the upper and lower interquartile; the median is represented as a horizontal line within the boxes; whiskers extend to the most extreme data point which is no more than 1.5 times the interquartile range. Different letter indicates statistical differences based on pairwise Mann–Whitney tests (two-sided, $p$-value < 0.05). NMDS stress: 0.10.

our study is a novel sulfur-oxidizing bacterium from the *Ca*. Thiodiazotropha genus that also showed high expression of sulfur oxidation genes. It is possible that when sulfate reduction was inhibited in previous studies[6,16,18], it disrupted the redox cycling of sulfur by impeding the flow of reduced sulfur for bacterial sulfur oxidation. Thus, not only was nitrogen fixation mediated by sulfate-reducing microorganisms inhibited, but also nitrogen fixation coupled to sulfur oxidation was likely impaired. Further evidence supporting the significance of the cycling of sulfur on nitrogen fixation is that members of the *Gammaproteobacteria* class and *Desulfobacterota* phylum contributed the majority of nitrogenase and oxidative/reductive *dsrAB* transcripts in the root compartment of *S. alterniflora*. Previous work employing DNA and RNA amplicons or short read 'omics analysis in seagrass and other coastal wetland plant species, showed nitrogen fixation genes and their expression to be affiliated with microorganisms closely related to

sulfate-reducing and sulfur-oxidizing bacteria[8–10]. However, very few genomes for these sulfur-cycling organisms were available and a genome-centric analysis had not yet been applied. Here, we present MAGs from diazotrophic sulfate-reducing and sulfur-oxidizing bacteria that highly express genes for both nitrogen fixation and dissimilatory sulfur reactions. Thus, we propose that both dissimilatory sulfur reduction and oxidation, and more importantly, the rapid redox cycling of sulfur, stimulate nitrogen fixation in the root environment of *S. alterniflora*. Since we show that coastal wetland plants and seagrasses assemble similar root microbiomes, and macrophyte activity boosts both nitrogen fixation and the cycling of sulfur, this is most likely a common phenomenon in the root zone of coastal marine plants worldwide[6,16,37,38].

In most coastal marine ecosystems, water-saturated sediments are often depleted in oxygen within the first few millimeter's depth[14].

Because coastal marine ecosystems are bathed in seawater containing high sulfate concentrations (28 mM), sulfate-reducing microorganisms often perform the terminal step of organic matter decomposition[14]. Unlike aerobic respiration, energy flow during sulfate reduction is decoupled from the carbon cycle, with most of the free energy conserved in reduced sulfur compounds[39]. The chemically stored energy is subsequently released by biotic and abiotic oxidation reactions at oxic-anoxic interfaces coupled to the reduction of electron acceptors (oxygen, nitrate, or oxidized metals)[39]. Surface sediments of vegetated coastal marine ecosystems experience rapid re-oxidation of most reduced sulfur compounds. This process is highly concentrated in the microaerophilic root zone, which serves as a hotspot for the reaction[40–42]. Our results showing that the root compartment of coastal marine ecosystems is highly enriched in microorganisms with the capability for both sulfur reduction and oxidation provide further evidence for rapid sulfur cycling in the root zone. Furthermore, to the best of our knowledge, this is the first study to employ a multi-omics approach to reconstruct genomes and determine gene expression of uncultivated sulfate-reducing and sulfur-oxidizing microorganisms living in the roots of coastal wetland plants. Our findings also reveal that genes that catalyze oxidative and reductive reactions in the sulfur cycle along with those of nitrogen fixation are highly transcribed in the root environment, particularly in the stressed *S. alterniflora* phenotype that thrives in a sulfidic environment[12]. This is consistent with previous studies of salt marsh and seagrass ecosystems, in which genes of sulfur oxidation pathways were shown to be highly expressed in the root compartment of *S. alterniflora* and *Zostera* seagrass spp., respectively[8,9]. Thus, we propose that in contrast to terrestrial ecosystems, coastal marine plants rely on the rapid cycling of sulfur in their root zone for the breakdown of organic matter and recycling of nutrients through sulfate reduction along with the re-oxidation of terminal electron acceptors by sulfur oxidation. Furthermore, we propose that rapid rates of both oxidative and reductive sulfur reactions represent a key mechanism to sustain high rates of energetically-expensive nitrogen fixation in the root zone of coastal wetlands and seagrass plants.

In this study, we reveal the genomes, phylogeny, function, and gene expression of chemolithoautotrophic symbionts discovered in the roots of *S. alterniflora*. Endosymbionts from the *Ca*. Thiodiazotropha genus were initially discovered living in symbiosis with lucinid clams, where they fix carbon and serve as a source of carbon and nitrogen to the animal host using energy gained from the oxidation of reduced forms of sulfur[22]. Recent studies using fluorescence in situ hybridization (FISH) microscopy and SSU rRNA gene metabarcoding have shown that *Ca*. Thiodiazotropha bacteria also inhabit the roots of a diverse array of seagrass species along with that of the coastal cordgrass *S. alterniflora*[12,21]. Here, we show that close relatives of diazotrophic sulfur chemosymbionts associated with lucinid clams are highly abundant and active in the roots of *S. alterniflora*. We report genomes and gene expression profiles from *Ca*. Thiodiazotropha retrieved from macrophyte root samples and show that our MAGs formed a monophyletic clade with those from previously described marine invertebrate animals. Similar to what has been reported in their symbiosis with lucinid clams, the *S. alterniflora* symbionts highly expressed genes for sulfur oxidation, carbon fixation, and nitrogen fixation[22,26]. However, the presence and expression of genes for glycolysis and the TCA cycle point to a mixotrophic lifestyle, akin to what is observed in their symbiosis with lucinid clams[22,43]. High expression of high-affinity oxygen reductases, such as the cytochrome c oxidase cbb3-type and cytochrome bd ubiquinol oxidase, may serve as adaptations for oxygen respiration under sulfidic conditions as well as an oxygen scavenging strategy for nitrogen fixation[44,45]. Moreover, the fluctuation of the redox state in the root zone of macrophytes during diel and tidal cycles may influence how *Ca*. Thiodiazotropha gspp partition aerobic and anaerobic metabolic processes[42,46,47]. Further

investigation is warranted to uncover the biogeography, host specificity, and temporal dynamics of *Ca*. Thiodiazotropha in association with macrophyte roots.

Studies of lucinid clams have shown that the *Ca*. Thiodiazotropha symbionts are horizontally transmitted[48]. Furthermore, invertebrate colonization by *Ca*. Thiodiazotropha symbionts is not restricted by either the host or symbiont species[26]. The flexibility of this symbiotic relationship could explain the evolution of *Ca*. Thiodiazotropha colonization in macrophyte roots. Previous studies have suggested that the root zone of seagrass species serves as a reservoir of chemosymbionts for marine invertebrates[28]. However, our studied marsh lacks lucinid clams or any marine invertebrate known to harbor bacterial chemosymbionts. Further application of a genome-centric approach to both macrophyte plant and lucinid clam hosts is needed to better interrogate if chemosymbiont populations are restricted by host biology at the kingdom level (i.e., to either plant or animal hosts), or if symbionts are shared and horizontally transferred between plant and invertebrate species. In contrast to what is observed in lucinid clams, where a single or few bacterial species dominates the gill microbiome; in *S. alterniflora* and seagrass species, *Ca*. Thiodiazotropha spp. do not outcompete other microbial species in the root compartment[26,27]. However, the symbionts comprise a large proportion of the microbial community, and are amongst the most active species[12].

We propose that the symbiosis between *S. alterniflora* and sulfur-oxidizing bacteria represents a key adaptation supporting the resilience of coastal salt marsh ecosystems to environmental perturbations. Intertidal wetland ecosystems are vulnerable to climate change because they are located in a narrow elevation range determined by tidal amplitude[49]. The Intergovernmental Panel on Climate Change (IPCC) has projected an increase in sea level between 0.38 m and 0.77 m by 2100 (IPCC-AR6[50]). Although coastal wetland ecosystems are dynamic and adapt to sea level rise by increasing sediment accretion rates, ecosystem models predict that a large area of present-day marsh will drown because of accelerated sea level rise[51]. Increased hydroperiods will impact the redox balance of vegetated sediments, imposing more severe anoxia and physiological stress from sulfide toxicity to wetland plants. Under this scenario, a symbiosis of coastal vegetated plants with sulfur-oxidizing bacteria could alleviate sulfide stress while at the same time coupling it to carbon and nitrogen fixation for potential plant uptake. However, the mechanism for carbon and/or nitrogen transfer between sulfur-oxidizing symbionts and the host plants still remains elusive and requires further research. Similar to Thomas et al.[8], we show the transcription of sulfur oxidation genes is greater in the stressed *S. alterniflora* short phenotype. In seagrass ecosystems, Martin et al.[52] also found that the seagrass root microbiome was more enriched in *Ca*. Thiodiazotropha spp. under stress conditions. Thus, we suggest that the *S. alterniflora* – *Ca*. Thiodiazotropha symbiosis is an adaptive interaction to anoxic soil conditions, whereby the host plant responds to stress from elevated dissolved sulfide concentrations. Further, we show that the short *S. alterniflora* phenotype, which harbors a greater nitrogenase transcript abundance of *Ca*. Thiodiazotropha chemosymbionts, also displayed the greatest rates of N fixation of all assessed marsh compartments. Stressed plants benefit from the symbiotic relationship through reduced sulfide toxicity and the coupling of sulfide oxidation to nitrogen fixation, with nitrogen likely transferred to the plant host. Conversely, the tall *S. alterniflora* phenotype showed a greater expression of genes involved in the internal cycling of nitrogen, which is consistent with previous studies showing greater rates of nitrogen mineralization in this zone of the ecosystem[12].

## Methods

### Study site, field sampling, and sample processing

All field sampling was performed within the United States in the state of Georgia. Sampling was performed with authorization from the

Georgia Department of Natural Resources (File: LOP20190067). Specifically, we sampled salt marsh ecosystems within the Georgia Coastal Ecosystem - Long Term Ecological Research (GCE-LTER) site 6, located on Sapelo Island, GA (Lat: 31.389° N, Long: 81.277° W). Four ~100 m transects along the tall to short *Spartina alterniflora* gradient were studied in July 2018, 2019, and 2020 (Fig. 1, further site description in ref. 12). A combination of multi-omics approaches and biogeochemical rate measurements were performed across different salt marsh compartments and phenotypes of *S. alterniflora*.

The microbiome of *S. alterniflora* was analyzed using shotgun metagenomics from three compartments: sediment, rhizosphere, and root. The tall and short *S. alterniflora* extremes of the four studied transects were sampled (n: 4 transects * 2 *S. alterniflora* phenotypes * 3 compartments = 24). Samples from transects 1 and 2 were collected in July 2018, while those from transects 3 and 4 were collected in July 2019. All sampling was performed at the 0–5 cm depth profile. Root-associated samples were washed two times with creek water in the field to remove coarse chunks of sediment attached to the plant. All samples were immediately flash-frozen in an ethanol and dry ice bath, and stored at −80 °C until DNA extraction.

Samples for metatranscriptomic analysis and biogeochemical rate measurements were collected in July 2020 only from transect 4 to reduce plant disturbance time before flash freezing and incubations, respectively. Five independent plants at least 3 meters apart were sampled with a shovel at the two *S. alterniflora* biomass extremes. Sampling was performed in the morning (~9:00 am) during low tide. A 25-cm diameter marsh section including several *S. alterniflora* shoots, undisturbed root system, and sediment was sampled up to at least 20 cm depth and transferred to a 5-gallon bucket. Plant samples were immediately transported to the field lab with no evidence of physiological stress noted after sampling. For each plant, a paired sample of sediment was collected. In the lab, roots were washed with creek water two times. After the first wash, roots with sediment attached were collected and defined as the rhizosphere + root compartment. Live, sediment-free roots were sampled after the second wash and defined as the root compartment. Top 5 cm from each sample were homogenized and used for total RNA extractions and measurements of nitrogen fixation rates. Root samples for RNA extractions were washed in an epiphyte removal buffer in ice, as in Simmons et al.[53]. Sediment and root samples for RNA analysis were immediately flash-frozen in an ethanol dry ice bath, and stored at −80 °C until extraction. Total root processing time since washing to flash freezing was about 1.5 h due to *S. alterniflora* intertwined root system with plant debris. Due to the limited sample amount, we did not prepare rhizosphere libraries for metatranscriptomic analysis.

### Nitrogen fixation rates

Rates of N fixation were calculated in $^{15}N_2$ incubations in 14 ml serum vials as in Leppänen et al.[54]. Rate measurements were started at the same day of plant collection, within 8 h after plant sampling. Rates were measured for all samples under both oxic and anoxic conditions. About 2 g and 1 g of wet weight was used for tracer gas incubations of sediment and root samples, respectively. An additional subsample of 2 g was immediately oven-dried at 60 °C for 72 h to be used as a pre-incubation control. In all vials containing root samples, 4 ml of autoclaved and 0.22 μm filter-sterilized artificial seawater was added to avoid tissue desiccation. No artificial seawater was added to sediment and rhizosphere samples since they were already water-saturated. In anoxic incubations, vials were flushed with $N_2$ gas for 5 min. After sealing all incubation vials, 2.8 ml of gas was removed and immediately replaced with 2.8 ml of $^{15}N_2$ (98% enriched, Cambridge Isotope Laboratories Inc, USA). Vials were overpressured by adding 0.3 ml of air or $N_2$ gas in oxic or anoxic incubations, respectively. The incubations were carried out in the dark and at room temperature for 24 h. At the end of the incubation, samples were oven-dried at 60 °C for 72 h,

and ground using a PowerGen high throughput homogenizer (Fisherbrand, Pittsburgh, PA). Ground samples were sent to the University of Georgia Center for Applied Isotope Studies (https://cais.uga.edu/) for carbon and nitrogen elemental analysis and $^{13}C$ and $^{15}N$ stable isotope analysis. Elemental analysis was performed by the micro-Dumas method, while isotopic analysis by isotope ratio mass spectrometry. Rates of $^{15}N$ incorporation were calculated per dry weight basis (DW) as in Leppänen et al.[54]:

$$\text{N uptake rate}\left(\text{nmol g}^{-1}\text{DWh}^{-1}\right) = \frac{\frac{1}{100} \times \frac{\%N}{100} \times \left[\frac{\text{atom}\%\text{sample} - \text{atom}\%\text{control}}{MW(N_2)}\right] \times 10^9 \times \frac{100}{\text{atom}\%\text{headspace}}}{\text{hours}}$$

(1)

Where %N is the N percent concentration of the oven-dried sample, and $MW(N_2)$ is the molecular weight of $N_2$ (28.013446).

### Nucleic acid extractions and multi-omics library preparation

For DNA extractions, compartment separation of the rhizosphere and root microbiomes was performed by sonication in an epiphyte removal buffer, as detailed in Simmons et al.[53]. Extracellular dissolved or sediment-adsorbed DNA was removed from bulk sediment samples according to the Lever et al.[55] procedure. DNA extractions from all samples were performed using the DNeasy PowerSoil kit (Qiagen, Valencia, CA) following the manufacturer's instructions. Shotgun metagenome sequencing was performed on an Illumina NovaSeq 6000 S4 2 × 150 Illumina flow cell at the Georgia Tech Sequencing Core (Atlanta, GA).

RNA from 4 biological replicates of both the sediment and root compartments from the tall and short phenotypes of *S. alterniflora* was extracted using the ZymoBIOMICS RNA Miniprep (Zymo Research Corp) kit according to the manufacturer's protocol (n: 4 replicates * 2 *S. alterniflora* phenotypes * 2 compartments = 16 samples). Rigorous DNA digestion was done with the TURBO DNase kit (Invitrogen), and eukaryotic mRNA was removed by binding and discarding the eukaryotic mRNA polyA region to oligo d(T)25 magnetic beads (England Biolabs). Finally, rRNA from both plant and prokaryotic organisms was depleted using the QIAseq FastSelect -rRNA Plant, and −5S/16 S/23 S kits, respectively (Qiagen, Valencia, CA). Metatranscriptomic libraries were sequenced in two lanes of Illumina's NovaSeq 6000 System flow cell utilizing the NovaSeq Xp workflow (SE 120 bp) at the Georgia Tech Sequencing Core (Atlanta, GA). We retrieved 203, and 415 Gpb of metagenomic and metatranscriptomic raw sequences, respectively; with a median sequencing effort of 7.8, and 25.3 Gbp per metagenomic and metatranscriptomic library (Supplementary Data File S1). After quality control and in silico removal of host and rRNA reads, median sequencing effort decreased to 5.2 and 8.3 Gbp per metagenomic and metatranscriptomic library, respectively (Supplementary Data File S1).

### Gene and transcript quantification of the prokaryotic 16S rRNA gene

Prokaryotic abundance and a proxy of prokaryotic activity were measured by quantitative polymerase chain reaction (qPCR) and reverse transcription-qPCR (RT-qPCR) of the SSU rRNA gene, respectively. All samples used for metagenome and metatranscriptome analysis were quantified by qPCR and RT-qPCR, respectively. Samples were analyzed in triplicate using the StepOnePlus platform (Applied Biosystems, Foster City, CA, USA) and PowerUp SYBR Green Master Mix (Applied Biosystems, Foster City, CA, USA). Reactions were performed in a final volume of 20 μl using the standard primer set for the prokaryotic SSU rRNA gene: 515 F (5′-GTGCCAGCMGCCGCGGTAA′) and 806 R (5′-GGACTACHVGGGTWTCTAAT′)[12,56]. To avoid plant plastid and mitochondrial DNA/cDNA amplification from root samples, peptide nucleic acid PCR blockers were added to all qPCR and RT-qPCR reactions at a concentration of 0.75 μM[57]. Standard calibration curves were performed using a 10-fold serial dilution ($10^3$ to $10^8$ molecules) of standard

pGEM-T Easy plasmids (Promega, Madison, WI, USA) containing target sequences from *Escherichia coli* K12. Melting curve analyses was used to check for PCR specificity. Prokaryotic gene and transcript abundance of the SSU rRNA gene were calculated as gene and transcript copy number g$^{-1}$ of fresh weight, respectively.

## Metagenomic and metatranscriptomic quality control

Metagenomic and metatranscriptomic raw reads were quality trimmed (quality phred score <20), and filtered for Illumina artifacts, PhiX, duplicates, optical duplicates, homopolymers, and heteropolymers using JGI's BBTools toolkit v.38.84[58]. Reads shorter than 75 bp were removed, and the quality of both metagenomic and metatranscriptomic libraries was assessed with FastQC[59]. Remaining reads were mapped against the only publicly available *S. alterniflora* genome (NCBI BioProject: PRJNA479677) with bowtie2 v.2.4.2[60], followed by removal of short reads that aligned to the *S. alterniflora* genome using samtools (parameters: view -u -f 12 -F256[61]). In addition, rRNA reads from metatranscriptomic samples were removed using sortMeRNA v.4.3.4[62]. Finally, DNA contamination in metatranscriptomic samples was assessed as indicated by Johnston et al.[63]. Short read transcripts were mapped to all assembled contigs, and strand-specificity (consistency in sense/antisense orientation) was calculated for all genes with more than 100 hits. All assessed samples had a greater than 95% average strand-specificity; thus, considered to be free of DNA contamination (Supplementary Data File S1). Filtered, quality-trimmed, and host-free reads were utilized for subsequent analyses.

## Metagenomic and metatranscriptomic short read functional analysis and nonpareil diversity

Short reads from all metagenomic and metatranscriptomic samples were aligned against the eggNOG protein database (release 5.0.2) using eggnog-mapper v.2.1.9[64]. The DIAMOND tabular outputs were filtered by retrieving only the best hit based on bitscore. Hits with less than 30% identity, less than 30% match of the read length, or that did not match a prokaryotic domain were removed from the analysis. Gene profiles for each metagenomic and metatranscriptomic library were constructed by counting hits of predicted KEGG orthology. Due to differences in sequencing effort and genome size between multi-omics libraries, the count matrixes were normalized by dividing them by their median count of 10 universal single-copy phylogenetic marker genes (K06942, K01889, K01887, K01875, K01883, K01869, K01873, K01409, K03106, and K03110) as in Salazar et al.[36]. Functional gene and transcript profiles were analyzed by principal coordinate analysis utilizing the Bray-Curtis dissimilarity distance. Further, for both metagenomic and metatranscriptomic profiles, multivariate variation of the Bray-Curtis dissimilarity matrix was partitioned to compartment (sediment, rhizosphere, and root) and *S. alterniflora* phenotype, based on a permutational multivariate analysis of variance (PERMANOVA) with 999 permutations performed in vegan v. 2.5.7[65]. The normalized relative abundance of selected terminal oxidases and genes/transcripts from the carbon, nitrogen, and sulfur cycles were assessed by compartment and *S. alterniflora* phenotype.

After removing reads annotated as Eukaryotic by eggnog-mapper, we calculated nonpareil diversity from all metagenomic samples using nonpareil v. 3.401[66]. Nonpareil diversity is a metric estimated from the redundancy of whole genome sequencing reads and has been shown to be closely related to classic metrics of microbial alpha diversity such as the Shannon index.

## Recovery of metagenome-assembled genomes (MAGs)

The following binning approach was motivated by a recently proposed method of iteratively subtracting reads mapping to MAGs, re-assembling, and re-binning metagenomic libraries to increase the number of recovered genomes[67]. An initial assembly using idba-ud v1.1.3 with pre-error-correction for highly uneven sequencing depth was performed

for all individual metagenomic libraries (default parameters), as well as co-assemblies grouping libraries from the same *S. alterniflora* phenotype and compartment (--mink 40 --maxk 120 --step 20 --min_contig 300)[68]. The resulting contigs from both individual and co-assemblies were binned using three different algorithms with default options: MaxBin v.2.2.7, MetaBAT v.2.15, and CONCOCT v1.1.0[69–71]. Recovered bins were dereplicated with DAS Tool v1.1.2[72], and the output was refined for putative contamination with MAGPurify v2.1.2[73]. Completeness, contamination, and quality score (Completeness – 5*Contamination) were calculated with MiGA v0.7[74]. Recovered MAGs with a quality score less than 50 were considered low quality and discarded. A second round of assembly and binning was performed after discarding short reads that mapped against MAGs. Metagenomic libraries were mapped against MAGs using bowtie2 v.2.4.2 with default parameters. Paired reads that mapped against MAGs were discarded using samtools v1.9 (parameters: view -F 2[61]). Filtered metagenomic libraries were co-assembled once again using idba-ud v1.1.3 with pre-error-correction for highly uneven sequencing depth (parameters: --mink 40 --maxk 120 --step 20 --min_contig 300) in four groups: (i) tall *S. alterniflora* root, (ii) short *S. alterniflora* root, (iii) tall *S. alterniflora* sediment and rhizosphere, and (iv) short *S. alterniflora* sediment and rhizosphere. Binning, MAGs' refinement, and quality control were performed as explained for the first iteration. Finally, MAGs from both iterations were grouped into genomospecies (gspp, singular gsp) by clustering MAGs with ANI > 95% using the MiGA v0.7 derep_wf workflow[74]. MAGs with the highest quality score were selected as representatives of their gsp, and most downstream analyses were performed with them.

Genomospecies relative abundance was estimated for each metagenomic library as in Rodriguez-R et al.[67]. Sequencing depth was calculated per position using bowtie2 v2.4.2 with default parameters[60] and bedtools genomecov v2.29.2 (parameters: -bga[75]). Bedtools output was truncated to keep only the central 80% values, and the mean of all retained positions was calculated using BedGraph.tad.rb from the enveomics collection, a metric defined as TAD$_{80}$ (truncated average sequencing depth)[76]. Relative abundance of each gsp was calculated by dividing TAD$_{80}$ by genome equivalents estimated for each metagenomic library with MicrobeCensus v1.1.0[77].

## MAG taxonomy, phylogenetic analysis, and gene functional annotation and expression

Taxonomic classification of all MAGs was performed by GTDB-Tk v2.1.0 using the reference database GTDB R07-RS207[34]. A maximum-likelihood phylogenetic tree of all gspp was constructed using a 400 universal marker genes database in PhyloPhlAn v3.0.58 (parameters: -d phylophlan, --msa mafft, --trim trimal, --map_dna diamond, --map_aa diamond, --tree1 iqtree, --tree2 raxml, --diversity high, --fast[78]). The phylogenetic tree was decorated and visualized in ggtree v2.0.4[79].

Protein-encoding genes from all binned MAGs were predicted with Prodigal v2.6.3 using default parameters[80], and resulting amino-acid sequences aligned against the eggNOG database (release 5.0.2) using eggnog-mapper v2.1.9[64]. The DIAMOND tabular output was filtered by retrieving only the best hit based on bitscore, and removing hits with less than 30% identity and/or less than 50% match length.

Metagenomic and metatranscriptomic short reads were mapped against functionally annotated ORFs of all gspp using megablast v2.10.1. Only hits with greater than 95% percent identity and 90% read alignment were retained. The roots from the short and tall phenotype had a median of 17.4% and 5.4% reads mapping back to the MAGs, respectively. Percent contribution of genes and transcripts to the total microbial community was assessed by dividing the number of hits from each gsp against the total number of functionally annotated short reads (from section "Recovery of metagenome-assembled genomes (MAGs)"). When assessing the transcript expression profile of a specific gsp, the number of mapped hits per annotated transcript was

normalized by dividing it by gene length (bp) and the median abundance of 10 universal single-copy phylogenetic marker genes of the prokaryotic community (K06942, K01889, K01887, K01875, K01883, K01869, K01873, K01409, K03106, and K03110).

Finally, since the phylogeny of the *dsrAB* gene allows to discriminate between the oxidative and reductive *dsrAB* types, we aligned all recovered genes from our binned gspp to a reference alignment[35] with Clustal Omega v1.2.4[81], and built an approximately-maximum-likelihood phylogenetic tree with FastTree v2.1.11[82]. Gene type was inferred based on placement in the phylogenetic tree.

### Non-binned scaffold analysis

Scaffold-level analysis from all assemblies and co-assemblies was conducted to enhance the representation of short reads mapping back to the nitrogenase and dissimilatory sulfite reductase genes. Prodigal v2.6.3, with default parameters, was used for predicting ORFs[80]. Functional annotation of predicted ORFs was carried out using eggnog-mapper v2.1.9 (database release 5.0.2, --pident 30, --query_-cover 50[64]). All ORFs annotated as *dsrA* or *nifK* were clustered at 95% nucleotide identity using cd-hit-est v4.8.1[83]. Metatranscriptomic short reads were aligned against the representatives from *dsrA* and *nifK* cd-hit-est clusters using megablast, similar to MAGs ORFs (section "MAGs taxonomy, phylogenetic analysis, and gene functional annotation and expression"). The taxonomy of ORFs not associated with MAGs was classified at the genus level using kaiju v1.10 against the NCBI-nr database (retrieved on May 10th, 2023)[84]. In cases where an ORF was not classified by kaiju, the higher taxonomic classification according to eggnog-mapper was used.

### Phylogenetic reconstruction of the Sedimenticolaceae family

Publicly available genomes from all species from the *Sedimenticolaceae* family, according to GTDB R07-RS207 and all *Candidatus* Thiodiazotropha genomes from the Osvatic et al.[25] study, were retrieved from NCBI. A full list and characteristics of genomes used for this analysis is found in Supplementary Data File S10. A phylogenetic tree of all retrieved genomes and binned genomes from this study was constructed in PhyloPhlAn v3.0.58 using 382 out of the PhyloPhlAn 400 universal marker genes database (parameters: -d phylophlan, --msa mafft, --trim trimal, --map_dna diamond, --map_aa diamond, --tree1 fasttree, --tree2 raxml, --diversity low, --accurate[78]). The phylogenetic tree was decorated and visualized using ggtree v2.0.4[79].

### Amplicon analysis of the 16S rRNA

RNA extractions performed for quantification of the 16S rRNA (section "Gene and transcript quantification of the prokaryotic 16S rRNA gene") were reverse transcribed using the SuperScript IV First-Strand Synthesis System kit (Invitrogen) following manufacturer instructions. Amplicon sequencing of the 16S rRNA was performed on cDNA as in Rolando et al.[12]. Amplicons were amplified for the 16S rRNA V4 region using primers 515 F (5'- GTGYCAGCMGCCGCGGTAA') and 806 R (5'-GGACTACNVGGGTWTCTAAT')[85,86]. Reactions were performed in 5-ng cDNA template in a solution containing DreamTaq buffer, 0.2 mM dNTPs, 0.5 μM of each primer, 1.25 U DreamTaq DNA polymerase, and 0.75 μM of each mitochondrial (mPNA) and plastid (pPNA) peptide nucleic acid (PNA) clamps to reduce plant plastid and mitochondrial cDNA amplification. Amplicons were sequenced on an Illumina MiSeq2000 platform using a 500-cycle v2 sequencing kit (250 paired-end reads) at Georgia Tech Sequencing Core (Atlanta, GA). Cutadapt v3.7 was employed to remove primers from raw fastq files[87]. We inferred Amplicon Sequence Variants (ASVs) from quality-filtered reads using DADA2 version 1.10[88]. Chimeric sequences were removed using the removeBimeraDenovo function in DADA2. Taxonomic assignment was performed utilizing the Ribosomal Database Project (RDP) Naive Bayesian Classifier[89] in conjunction with the SILVA SSU rRNA reference alignment (Release 138[90]). The relative abundance of the *Sedimenticolaceae* and *Desulfosarcinaceae* families, as well as *Ca.* Thiodiazotropha and *Desulfatitalea* genera was estimated by plant compartment and *S. alterniflora* phenotype.

### Analysis of root microbiomes from contrasting ecosystems

Publicly available 16S rRNA gene amplicon datasets, generated from next-generation sequencing, were used to characterize the community assembly of root microbiomes from coastal marine ecosystems. Studies were selected based on google scholar queries using a combination of the following keywords: "salt marsh", "coastal", "wetland", "mangrove", "seagrass", "seabed", "crop", "bog", "plant", "root", "endosphere", "microbial community", "microbiome", "amplicon", "next-generation sequencing", "16S rRNA", and "SSU rRNA", as well as based on the authors' prior knowledge. Only studies that collected environmental samples were included (i.e., no greenhouse or plants grown on potting media were included). When available, paired soil/sediment and rhizosphere samples were also retrieved. Twenty-two studies that met our requirements were selected, collecting a total of 2911 amplicon samples, with 1182 of them being from the root compartment across 56 different plant species. Selected plants were categorized into 4 different ecosystem types: seagrass meadows, coastal wetlands, freshwater wetlands, and other terrestrial ecosystems. A complete list of selected amplicon samples with accompanying metadata is available in Supplementary Data File S8.

Cutadapt v3.7 was used to detect and remove primer sequences from all datasets[87]. Primer-free sequences were quality filtered using DADA2's filterAndTrim function [options: truncLen = c(175,150), maxN = 0, maxEE = c(2,2), truncQ = 10, rm.phix = TRUE][88]. Trimmed reads were randomly subsampled to a maximum of 20,000 reads using seqtk v.1.3[91] and used as input for dada2 amplicon sequence variant (ASV) calling[88]. Chimeras were removed using the removeBimeraDenovo function from the DADA2 package. Taxonomy was assigned to ASVs utilizing the Ribosomal Database Project (RDP) Naive Bayesian Classifier[89] against the SILVA SSU rRNA reference alignment [Release 138[90]]. Sequences classified as chloroplast, mitochondrial, and eukaryotic or that did not match any taxonomic phylum were excluded from the dataset. Samples that had less than 5000 reads were removed at this stage. After all quality filtering steps, we kept on average 13,157 reads from the initial 20,000 subsampled reads per sample (65.8% reads). In order to merge studies from different sequence runs and primer sets, we grouped ASVs at the genus level before merging all studies into a single dataset in phyloseq v1.36[92]. All selected studies targeted the 16S rRNA gene V3–V4 region, with 55% of them using the 515 F/806 R primer set.

Microbial community assembly of the root microbiome was analyzed by performing a non-metric multidimensional scaling (NMDS) ordination utilizing the Bray-Curtis dissimilarity distance. Multivariate variation of the Bray-Curtis dissimilarity matrix was partitioned to ecosystem type, and compartment (soil/sediment, rhizosphere, and root), based on a PERMANOVA with 999 permutations performed in vegan v2.5.7[65]. Finally, putative function for sulfate reduction and sulfur oxidation was assigned based on taxonomic identity at the genus level as in Rolando et al.[12]. Taxa with known sulfate-reducing and sulfur-oxidizing capability is found in Supplementary Data File S9.

### Reporting summary

Further information on research design is available in the Nature Portfolio Reporting Summary linked to this article.

## Data availability

The raw metagenomic and metatranscriptomic sequences generated in this study have been deposited in the BioProject database (http://ncbi.nlm.nih.gov/bioproject) under accession codes PRJNA703972 and PRJNA950121, respectively. The metagenome-assembled genomes generated in this study have been deposited in the BioProject database

under accession code PRJNA703972. The amplicon 16S rRNA raw reads generated in this study have been deposited in the BioProject database under accession code PRJNA1034039. The nitrogen fixation rates generated in this study have been deposited in a Zenodo repository under accession code 7883423[93]. All accompanying metadata generated in this study are provided in the Supplementary Data Files and the Zenodo repository[93].

## Code availability

Custom scripts used in the present study are publicly available in a Zenodo repository: https://doi.org/10.5281/zenodo.7883423.

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

## Acknowledgements

This work was supported in part by an institutional grant (NA18OAR4170084) to the Georgia Sea Grant College Program from the National Sea Grant Office, National Oceanic and Atmospheric Administration, US Department of Commerce, and by a grant from the National Science Foundation (DEB 1754756). Any opinions, findings, and conclusions or recommendations expressed in this material are those of the authors and do not necessarily reflect the views of the National Science Foundation.

## Author contributions

Conceptualization: J.L.R., M.K., J.T.M., and J.E.K. Field sampling and analysis: J.L.R., M.K., T.S., Y.L., and J.E.K. Bioinformatic analysis: J.L.R., M.K., P.P., R.C., and K.T.K. Funding acquisition: J.E.K. Writing—original draft: J.L.R. with inputs from all authors.

## Competing interests

The authors declare no competing interests.
