## [Peer Review File · Nature Communications]

Sulfur oxidation and reduction are coupled to nitrogen fixation in the roots of the salt marsh foundation plant *Spartina alterniflora*Reviewer #1 (Remarks to the Author):

This work in general, is an exciting advance in plant-microbe symbioses, especially in marine systems. However, it is fairly narrow in scope as written. How broadly applicable could sulfur-oxidation be tied to N-fixation in marine systems? To the root zone of plants? I would like to see more linkages to those ideas more than what is currently in the manuscript. Other general thoughts – I would think that N fix/Sulfur oxidation is occurring more in the short plants just because the conditions are more suited (higher sulfide levels, less oxygen availability) and not because of the stress-gradient hypothesis mentioned in the manuscript. I also have several comments and questions that should be addressed during any revisions of this work.

Line 85. Please consider citing: König, S., Gros, O., Heiden, S. et al. Nitrogen fixation in a chemoautotrophic lucinid symbiosis. *Nat Microbiol* 2, 16193 (2017). <https://doi.org/10.1038/nmicrobiol.2016.193>; since this was published at the same time and in the same journal as the one listed.

Line 137. What portion of the communities/prokaryotic reads are these? Do all contain a 16S rRNA gene, and how did you determine the quality?

Line 150 and elsewhere. mark on the figure all taxa (MAGs) you mention here in the text, or at least the ones mentioned most frequently (Ca. Thiodiazotropha, Desulfosarcinaceae).

Line 171, In order for nitrification, there must be nitrogen (nitrate/nitrite) in the system. Were N species (at least nitrate, nitrite, and ammonium) concentrations measured in the sediments or other samples, for each condition?

Line 190-192. Of all transcripts mapped to MAGs, or to all ORFs/CDSs or even reads? Did you try mapping to any reference genomes or with less stringent conditions?

Line 212. How many genes were actually in the alignment and how much coverage was there of each MAG/gene? I believe PhyloPlhAn uses all genes present in any of the genomes examined unless you set up different conditions for clade-specific phylogenomics. Therefore, there may be much less than 400 genes that are relevant to your taxonomic profiles.

Line 237 paragraph. According to the FISH images, there is a patchy distribution within the root zone of Ca. Thiodiazotropha. This makes me question a few things: how variable is the abundance of different MAGs in this genus between metagenomes/between plants, etc.? How much co-occurrence is there and how might co-assembly confuse what is actually in the samples? Is the nitrogenase gene (or others) different enough between the species to capture differences in gene expression?

Line 277-278. What type of program was used? At least give the reader a bit more information here.

Line 293/349. Many, if not all, of the Ca. Thiodiazotropha spp. have the capacity to import sugars for their metabolism. There are abundant carbon compounds in roots/root exudates. Was there any evidence of mixotrophy? Or gene expression of other C update genes or mechanisms to gain E?

Line 298 and elsewhere. I am a firm believer of not directly referring to figures and tables in the discussion. It detracts and adds unnecessary words to the manuscript.

Line 301. Other studies? Citations are missing.

Line 303-305. Did you or others measure the amount of sulfate or sulfur compounds in these experiments? This would make for much stronger evidence for this statement. As it is, you don't have much evidence.

Line 306. But, you could measure sulfide or even other sulfur compounds (thiosulfate, etc.) depletion in the system? There may be ways to inhibit sulfide oxidation as well.

Lines 324, 341, and elsewhere. This amount of detail is not needed in a discussion section (giving concentrations).

Line 341. Not that much more. Have you thought to just map reads to the MAGs and normalize against the MAG abundance in each condition?

Line 371-373. Please do not present results in the discussion without first presenting them in the results section. I don't believe this result was in the results section.

Line 376. This is not always true, please check Osvatic, et al., 2023.

Line 379-380. You do not know this, at least the active portion – not in your supplemental figure, at least.

Line 394-395. Citations are needed.

Line 400-410. S oxidation under anoxic conditions - did you look for denitrification instead of oxygenic respiration? Not all *Ca; Thiodizotropha* spp. have that capability, I believe. That reminds me – since nitrogenase is sensitive to oxygen, what mechanisms do/could the microbes use to remove that effect during N fixation? For instance, is there any evidence for more N fixation at night when the plants are not producing oxygen? Are the stressed plants fixing more nitrogen just because they are in an environment more conducive to N fixation?

Line 411, elsewhere. Any studies of other seagrasses? Especially those in environments where there are Lucinidae?

Line 411-415. Please do not present results in the discussion without first presenting them in the results section. At least some of this was also not found in the results, I believe.

Line 445-459. Time of collection? Tidal influences? Amount of time between collection and actually freezing for RNA extraction? That seems to be very long. A lot can happen in that time.

Line 461. How long after collection? How were samples stored? What portion of roots were tested? Did you test multiple regions of the roots? Were there differences in the root structure between the two plant types?

What percentage of reads were identified as bacteria/archaea from each metagenome? What percentage of reads were actually binned?

Line 577-580. Idba-ud is generally a poor choice for assembly of metagenomes. Did you try other metagenome assemblers, such as Spade? Co-assemblies do a very poor job of assembling true MAGs especially with closely related species/strains found in the different samples, You likely get a lot of artificial recombination events happening. I do not recommend AT ALL. How many samples/MAGs were generated with co-assemblies?

Line 600. There are other estimates of genome coverage in metagenomes; have you tried CoverM?

Line 620-621. % of reads kept/used/identified per metagenome?

Line 625-628. Why not map to the reference genome and just use DESeq2 or edgeR or other method

to determine differential gene expression?

Line 668-669. But mixing primer sets at that level might lead to erroneous conclusions - some regions might resolve at genus or species level, others family or order levels.

Line 675-677. Please provide a bit more detail here.

Reviewer #2 (Remarks to the Author):

The authors have performed a very interesting study, executed in a solid way and written nicely and clear. The study builds well on previous work done by the authors providing novel insight in how sulfide stress is handled by a foundation plant through the interaction with sulfur cycling bacteria that also seem to have the additional benefit for the host that they fix nitrogen. The study is descriptive but uses contrasting conditions in a field situation to demonstrate the interaction. Experimental proof could be obtained by reciprocal transplantation experiments, seedling work as well as by manipulation of sulfide conditions.

The work is novel, the host-sulfur cycling microbes is mainly known from the deepsea and animal hosts. This study together with some previous work on seagrasses strongly suggest that they also play an important role in plants living at the interface with the sea. The combination of approaches combined in this work is impressive with sufficient but limited replication.

The work supports the conclusions and claims as presented. The used methodology is sound, though some chosen thresholds are arguable. For example, the MAGs used in this study are considered high quality at a completeness of 50% while generally this is so for 90%. It is unclear to what extent the story remains standing if the general threshold is used. I think the authors need to demonstrate the (lack of) effect of this threshold for the story of the paper.

In the performed Mann-Whitney test p-values <0.5 are considered significant, is there a typo here or is there a sound argument to use this threshold?

The method section is clear and provides enough details so that the work can be reproduced.

For detailed feedback and small comments and remarks, please see the attached pdf.

It seems highly likely that the process identified here is an important driver in the zonation of marine macrophytes which form very important ecosystems across the globe. This paper will certainly (re)activate the scientific community active in these ecosystems and hopefully will unite research across kingdoms to address the evolution of these interactions.

kind regards
Aschwin Engelen

Reviewer #3 (Remarks to the Author):

The submitted manuscript " Sulfur oxidation and reduction are coupled to nitrogen fixation in the roots of a salt marsh foundation plant species " explores the coupling of S cycling and N fixation during coastal wetland plant-microbe interactions, using tools from molecular microbial ecology and biogeochemistry. The study is ecologically intriguing, but before I can recommend its publication, I believe several improvements are necessary:

1. Streamlining and focusing the manuscript: The authors should seriously streamline the manuscript by focusing on the key story they want to convey. Currently, there are multiple stories being told simultaneously. For instance, in the introduction, the authors delve into the stress gradient hypothesis,

which might not be central to their study as no gradients were sampled, only two extremes along a hypothesized gradient. Additionally, aspects on C fixation are included in the introduction and discussion, which seem highly speculative based on the data. The meta-analysis comparing microbial communities across ecosystem types, while interesting, appears unnecessary for delivering the main story and could be used to prepare a separate manuscript. The main manuscript includes 6 figures, and there are another 10 figures in the supplement, many of which have more than 3 panels, further contributing to the manuscript's complexity.

2. Addressing novelty of findings: The manuscript's novelty is not clearly demonstrated due to the many aspects touched upon. The authors need to address specific concerns about the novelty of their findings. You state that previous studies have shown that *Ca. Thiodiazotropha* bacteria are not just found in lucinid clams but also inhabit the roots of a diverse array of seagrass species along with that of the coastal cordgrass *S. alterniflora* (Martin et al., 2020a; Rolando et al., 2022). Also, your statement in L317-321 reads like other studies already captured the key findings of your study: "Similar results have been obtained (..) in seagrass and other coastal wetland plant species, where nitrogen fixation genes and their expression have been affiliated with microorganism closely related to sulfate-reducing and sulfur oxidizing bacteria (Thomas et al., 2014; Crump, et al. 2018; Kolton et al., 2020)". The authors should clarify the ecological novelty of their results and emphasize how they contribute to the existing body of knowledge.

3. Providing evidence for stressed phenotype: Throughout the manuscript, the authors refer to a stressed phenotype of *S. alterniflora*, but no evidence is provided to support this claim. Physiological measurements are needed to back this notion. Isn't it possible that the short phenotype is well-adapted to its microenvironment and not necessarily more stressed than the tall phenotype? Additionally, referring to the study as a "gradient study" might be misleading since only two positions along a transect were sampled.

4. Strengthening evidence for coupling of S oxidation and N fixation: To provide stronger evidence for the coupling of sulfide oxidation and N fixation, the authors should address the following points:

- o Correlation across the entire microbial community: The manuscript currently shows the correlation between functional gene transcription for sulfur oxidation and nitrogen fixation within a single microbial species. The authors should consider exploring correlations across the entire microbial community.
- o Similar N fixation rates between phenotypes: The authors need to explain why N fixation rates do not significantly differ between *S. alterniflora* phenotypes. Additionally, Figure 2b presents root vs. sediment but not the rhizosphere (unlike panel a). The authors should clarify this discrepancy.
- o Lack of $\delta^{15}\text{N}$ figure: Given that the study primarily focuses on N fixation, the absence of a figure on natural abundance $\delta^{15}\text{N}$ of plant tissue is surprising. Substantial N_2 fixation by root microbes could be reflected in $\delta^{15}\text{N}$ (e.g. Högberg 1997 *New Phytol*). Likewise, differences in N_2 fixation between the two phenotypes should be reflected in the natural abundance of ^{15}N in plant tissues.
- o Consideration of sulfide oxidation as a measurable process: Contrary to the authors' claim, sulfide oxidation is not an "unmeasurable process" as suggested in Findlay et al. 2020 (<https://doi.org/10.1016/j.gca.2020.04.007>). This means, it would have been possible to provide more biogeochemical evidence for the coupling of N fixation and sulfide oxidation.

ITEMIZED RESPONSE TO REVIEWER COMMENTS

Reviewer #1 (Remarks to the Author):

Reviewer 1, comment 1:

This work in general, is an exciting advance in plant-microbe symbioses, especially in marine systems. However, it is fairly narrow in scope as written. How broadly applicable could sulfur-oxidation be tied to N-fixation in marine systems? To the root zone of plants? I would like to see more linkages to those ideas more than what is currently in the manuscript.

Reply: We appreciate the overall positive and constructive comments from the reviewer on our manuscript. We show that sulfur oxidation is coupled to nitrogen fixation in the roots of salt marsh plant *Spartina alterniflora*. However, we argue that this may be a common phenomenon in global coastal marine plants, ranging from mangroves and salt marshes to seagrass meadows. We broaden the scope of the manuscript in that direction, especially in the introduction and discussion sections. For example, please see lines 208-216, 542-545, 590-594. Please note that all referenced lines in the reply letter are linked to the track-changes PDF document.

Reviewer 1, comment 2:

Other general thoughts – I would think that N fix/Sulfur oxidation is occurring more in the short plants just because the conditions are more suited (higher sulfide levels, less oxygen availability) and not because of the stress-gradient hypothesis mentioned in the manuscript. I also have several comments and questions that should be addressed during any revisions of this work.

Reply: We agree and acknowledge that the right biogeochemical gradients have to be in place to stimulate sulfur oxidation coupled to N fixation. Please see lines 200-202. Also, in response to similar comments by Reviewers 1 and 3, we have decided to remove reference to the stress gradient hypothesis. Although we sampled the extremes of the gradient, we investigate the effect of environmental stress under these two contrasting conditions. The stress gradient concept has been well established for *Spartina* productivity adjacent to large tidal creeks in coastal marshes (see Mendelssohn and Morris, 2000).

Reviewer 1, comment 3:

Line 85. Please consider citing: König, S., Gros, O., Heiden, S. et al. Nitrogen fixation in a chemoautotrophic lucinid symbiosis. Nat Microbiol 2, 16193 (2017). <https://doi.org/10.1038/nmicrobiol.2016.193>; since this was published at the same time and in the same journal as the one listed.

Reply: Thanks, we agree. We cited the König et al. paper in line 130.

Reviewer 1, comment 3:

Line 137. What portion of the communities/prokaryotic reads are these? Do all contain a 16S rRNA gene, and how did you determine the quality?

Reply: Due to high diversity in the case of the sediment and rhizosphere compartments, and high-proportion of eukaryotic reads in the case of the root compartment, not a high percentage of reads mapped back to our MAGs. Percent of short reads mapping to scaffolds and MAGs was added to Supplementary Table S1. In the case of the root compartments, where we use a genome-centric approach, the median percent mapping back to MAGs was 6% and 22% in the tall and short phenotypes, respectively. This suggests that we mainly recovered the most abundant species. The above information as well as a call to further interrogate the microbial diversity of coastal wetland ecosystems can be found between lines 265-268. Presence/absence of the 16S rRNA gene was added to Supplementary Table S2. 68 out of the 239 MAGs had the 16S rRNA gene. Quality was assessed by using the quality score (Completeness – 5*Contamination). This was also added to the text between lines 262-264.

Reviewer 1, comment 4:

Line 150 and elsewhere. mark on the figure all taxa (MAGs) you mention here in the text, or at least the ones mentioned most frequently (Ca. Thiodiazotropha, Desulfosarcinaceae).

Reply: Thank you for the suggestion. The figure was updated to show the mentioned MAGs (please see new Figure 3)

Reviewer 1, comment 5:

Line 171, In order for nitrification, there must be nitrogen (nitrate/nitrite) in the system. Were N species (at least nitrate, nitrite, and ammonium) concentrations measured in the sediments or other samples, for each condition?

Reply: We decided to remove the nitrification rates from the manuscript in order to streamline the manuscript as suggested by Reviewer 3. We are focusing on the coupling of N fixation with sulfur oxidation and sulfate reduction. Nonetheless, we observe high concentrations of ammonium in the porewater of this marsh that would support nitrification (~100 µM on average in the porewater, see Supp Fig S3 in Rolando et al., 2022 which studied the same transects in Sapelo Island, GA). In the same publication, we show low nitrate porewater concentration (almost undetected in the short phenotype and < 12 µM for all tall *Spartina* samples).

Reviewer 1, comment 6:

Line 190-192. Of all annotated short reads to MAGs, or to all ORFs/CDSs or even reads? Did you try mapping to any reference genomes or with less stringent conditions?

Reply: The percentage is from reads mapping to MAGs to the total nifK and rdsrA functionally-annotated short reads. We have slightly modified the text to make it clearer (please see lines 339-342). We used stringent conditions (95% percent identity and 90% alignment) to make sure we were only mapping back to one genomospecies. For instance, median percent identity of nifK genes between different *Thiodiazotropha* spp is 86.4%., and the maximum similarity 90.7% identity.

Reviewer 1, comment 7:

Line 212. How many genes were actually in the alignment and how much coverage was there of each MAG/gene? I believe PhyloPlhAn uses all genes present in any of the genomes examined unless you set up different conditions for clade-specific phylogenomics. Therefore, there may be much less than 400 genes that are relevant to your taxonomic profiles.

Reply: According to the phylophlan manual <https://github.com/biobakery/phylophlan/wiki>, the default option is that genomes with less than 100 marker genes detected are discarded from the phylogenetic tree. Also, marker genes detected in less than 4 input genomes are discarded. In our case, it is true that not all genomes had the 400 marker genes, but using the default conditions, only 18 marker genes were discarded, keeping 382 marker genes for analysis. The concatenated alignment is trimmed for columns containing more than 65% gaps. In our case, the trimmed concatenated alignment used for the *Sedimenticolaceae* tree contained 14,693 amino-acid positions. We added that information in lines 372-375. The phylophlan options used for making the phylogenetic tree are found in lines 1038-1042.

Reviewer 1, comment 8:

Line 237 paragraph. According to the FISH images, there is a patchy distribution within the root zone of *Ca. Thiodiazotropha*. This makes me question a few things: how variable is the abundance of different MAGs in this genus between metagenomes/between plants, etc.? How much co-occurrence is there and how might co-assembly confuse what is actually in the samples? Is the nitrogenase gene (or others) different enough between the species to capture differences in gene expression?

Reply: The way we defined genomospecies (ANI > 95%) is intended to separate distinct biological units in a way that allow us to capture differences in abundance, gene content and expression of closely related species within the same sample. For this same reason, we set a stringent mapping threshold (95% percent identity and 90% alignment length) to quantify gene expression. After the stringent mapping, only

4.3% hits mapped to more than one gsp. Then, we removed the duplicated count from the gsp with the lowest percent identity (i.e., only best matches were used). Only 2.0% reads mapped to more than one gsp with the same percent identity value. These are highly conserved domains within proteins.

For the nitrogenase gene nifK, for instance, the median percent identity between different Thiodiazotropha gssp., was 86.4% (maximum: 90.7%, minimum: 83.8%), so well below our 95% nucleotide identity threshold to distinguish different versions (and species carrying them). We did find co-occurrence of Thiodiazotropha gssp. within the same plant species, which is also consistent with what has been found in amplicon studies (Rolando et al., 2022 and Martin et al., 2020). However, we were able to separate the gene expression between the different genomospecies (For instance see panel c from Fig. 4).

Reviewer 1, comment 9:

Line 277-278. What type of program was used? At least give the reader a bit more information here.

Reply: Putative function was inferred based on homology at the genus level with prokaryotes with known sulfur oxidation or sulfate reduction capability. The taxonomy list used with appropriate references was included in the supplementary information for greater transparency (please see Supplementary Table S11).

Reviewer 1, comment 10:

Line 293/349. Many, if not all, of the Ca. Thiodiazotropha spp. have the capacity to import sugars for their metabolism. There are abundant carbon compounds in roots/root exudates. Was there any evidence of mixotrophy? Or gene expression of other C update genes or mechanisms to gain E?

Reply: Thank you for this insight! Indeed, evidence for mixotrophy was found based upon further examination in response to the reviewer's comment. Our MAGs contained genes for glycolysis and the TCA cycle, and both were expressed in the assessed MAGs. We added this to both the Results and Discussion sections (see lines 397-399, and lines 677-679, respectively).

Reviewer 1, comment 11:

Line 298 and elsewhere. I am a firm believer of not directly referring to figures and tables in the discussion. It detracts and adds unnecessary words to the manuscript.

Reply: We appreciate the suggestion. We have removed references to figures and tables in the Discussion section.

Reviewer 1, comment 12:

Line 301. Other studies? Citations are missing.

Reply: Citations were added. Please see lines 556-557.

Reviewer 1, comment 13:

Line 303-305. Did you or others measure the amount of sulfate or sulfur compounds in these experiments? This would make for much stronger evidence for this statement. As it is, you don't have much evidence.

Reply: Neither we nor previous studies measured the evolution of sulfate or other sulfur compounds. Previous studies measured the difference in nitrogen fixation with and without molybdate inhibition. Due to lack of evidence, we have toned down our statements. Please see lines 556-578.

Reviewer 1, comment 14:

Line 306. But, you could measure sulfide or even other sulfur compounds (thiosulfate, etc.) depletion in the system? There may be ways to inhibit sulfide oxidation as well.

Reply: Due to the complex redox state of sulfur compounds, as well as isotopic exchange between S compounds, it is difficult to quantify S oxidation *in situ*. And it is even more difficult to partition differences in sulfur chemistry between compartments in the marsh. A published study referenced by Reviewer 3 measured 5 radiolabeled S pools to quantify sulfide oxidation to sulfate using intact cores. Despite their careful measurements using this complex method, thiosulfate and elemental sulfur oxidation as well as the oxidation of sulfide to intermediate oxidation states of S were not able to be quantified. Nonetheless, we have removed the relevant line from the manuscript, in response to the reviewer's comment.

Reviewer 1, comment 15:

Lines 324, 341, and elsewhere. This amount of detail is not needed in a discussion section (giving concentrations).

Reply: We removed concentration values in the discussion section.

Reviewer 1, comment 16:

Line 341. Not that much more. Have you thought to just map reads to the MAGs and normalize against the MAG abundance in each condition?

Reply: Due to low recovery of short reads back to MAGs, we decided to work with the short reads for ecosystem level responses. Short read analysis is not affected by issues with assembly due to high diversity. Furthermore, in some cases MAGs are only present

in one of the two *Spartina* phenotypes or in one of the three compartments. In many cases, MAG abundances can be very low, which impedes the analysis proposed by the reviewer. In Figure 4b we effectively show that both the oxidative and reductive *dsrA* transcripts are significantly more abundant in the short phenotype than the tall in the root compartment. In general, the genes for dissimilatory sulfate reduction (reductive *dsrAB* and *aprAB*) were among the most transcribed genes in the root compartment (figure Supplementary Figure S1).

Reviewer 1, comment 17:

Line 371-373. Please do not present results in the discussion without first presenting them in the results section. I don't believe this result was in the results section.

Reply: Thanks for the comment. We moved that sentence to the results section (see lines 392-394)

Reviewer 1, comment 18:

Line 376. This is not always true, please check Osvatic, et al., 2023.

Reply: Thanks for the correction. We have modified the text accordingly. Still, there are few species colonizing the gill microbiome, compared to the more diverse root microbiome. We modified the text. Please see lines 711-714.

Reviewer 1, comment 19:

Line 379-380. You do not know this, at least the active portion – not in your supplemental figure, at least.

Reply: We included Supplementary Figure S6 showing the top 15 genomospecies with greater transcript abundance per metatranscriptomic sample in the roots of *Spartina alterniflora*. A *Thiodiazotropha gsp* was the one with the greatest transcript counts and 5 species from that genus appear as species along the highest transcript abundance.

Reviewer 1, comment 20:

Line 394-395. Citations are needed.

Reply: Previous lines 394-395 referenced to the fact that the transfer mechanism -if any- by which fixed nitrogen and carbon is passed from *Thiodiazotropha spp.* to the plant is still unknown. We did not add citations since there are no studies showing it yet.

Reviewer 1, comment 21:

Line 400-410. S oxidation under anoxic conditions - did you look for denitrification instead

of oxygenic respiration? Not all Ca; Thiodiazotropha spp. have that capability, I believe. That reminds me – since nitrogenase is sensitive to oxygen, what mechanisms do/could the microbes use to remove that effect during N fixation? For instance, is there any evidence for more N fixation at night when the plants are not producing oxygen? Are the stressed plants fixing more nitrogen just because they are in an environment more conducive to N fixation?

Reply: Only two Thiodiazotropha gsp had genes for nitrate reduction (gsp 32, gsp 33). However, both cytochrome c oxidase cbb3-type and the cytochrome bd ubiquinol oxidase (which use O₂ as terminal electron acceptor under microaerophilic conditions) were more transcribed than nitrate reduction (Supplementary Table S6). This information was added in lines 397-399, and 404-406.

It is known that other prokaryotic species that fix nitrogen under aerobic conditions use O₂ scavenging mechanisms such as the cytochrome bd ubiquinol oxidase and the cytochrome c oxidase cbb3-type to deplete O₂ for N fixation. This could be the case in our studied microorganisms as well. Further, besides these two oxidases, Thiodiazotropha gsp also contain the coxBAC operon for the COX catalytic core. Meaning, they have 3 different cytochrome c oxidases that could be used under different O₂ concentrations. However, in the two most active Thiodiazotropha gsp., genes for microaerophilic O₂ respiration were more transcribed than the coxBAC operon. The partitioning of N fixation during light/dark cycles is interesting, and requires further research, but beyond the scope of this manuscript. It is known that the root zone of plants is more oxidized during the day when plants actively perform gas exchange, and more reduced at night when they close stomata. This could explain differences in N fixation following diel cycles. We have added an explanation in lines 679-700.

Reviewer 1, comment 22:

Line 411, elsewhere. Any studies of other seagrasses? Especially those in environments where there are Lucinidae?

Reply: Previous line 411 was removed, along that whole paragraph to streamline the manuscript. However, a few references on seagrasses were added to the manuscript (Capone 1982, Cúcio et al. 2016, Cúcio et al. 2018, Fahimipour et al, 2017).

Reviewer 1, comment 23:

Line 411-415. Please do not present results in the discussion without first presenting them in the results section. At least some of this was also not found in the results, I believe.

Reply: This section of the discussion was removed in order to streamline the manuscript nitrogen fixation coupled to sulfur oxidation in the root zone.

Reviewer 1, comment 24:

Line 445-459. Time of collection? Tidal influences? Amount of time between collection and actually freezing for RNA extraction? That seems to be very long. A lot can happen in that time.

Reply: Plants were collected in the morning (around 9 am) during low tide. A large section of marsh sediment with multiple *Spartina* shoots was transferred to the lab in 5-gallon buckets (marsh section: diameter 25 cm, depth: at least 20 cm). Plants did not show evidence of stress after transplantation to the bucket and transport to the field lab. The field lab is 10 minutes away from the sampling location. Plant processing started within one hour of sampling. Once root washing started, the processing was performed as fast as possible. Washing roots in salt marsh sediment can be difficult due to a dense mat of mixed *Spartina* root and decomposing roots in the sediment. We wanted to make sure we were collecting roots connected to living stems. Roots were flash frozen within 1.5 hours after processing started, which included the removal of rhizosphere material in which samples were kept on ice. The high degree of replication between biological samples and clear differences between compartments and *Spartina* phenotypes indicate that the transcripts were not highly impacted by processing. However, it is certainly possible that sampling and processing procedures introduced biases to our metatranscriptomes -as in any other environmental RNA study studying compartments complicated to separate. Thus, all this information was added to the methods section between lines 788-825, and 832-835 for future discussion of our findings.

Reviewer 1, comment 25:

Line 461. How long after collection? How were samples stored? What portion of roots were tested? Did you test multiple regions of the roots? Were there differences in the root structure between the two plant types?

Reply: Yes, we tested a combination of root tissue with different morphologies. This was done the same day the plants were collected -within 8 hours after plant sampling-. This was added to the methods. Please see lines 838-839. There were no clear differences in root morphology between the short and tall phenotype. Although we did not measure it, it was our impression that plants from the short phenotype had a greater root-to-shoot ratio.

Reviewer 1, comment 26:

What percentage of reads were identified as bacteria/archaea from each metagenome? What percentage of reads were actually binned?

Reply: Both can be found in Supplementary Table S1. Please see previous response for a more comprehensive description of reads mapping back to MAGs. As for the

percentage of reads identified as bacteria/archaea, we took the following approach. First, we removed reads that mapped to a draft genome of *Spartina alterniflora*. Then, we also removed all eukaryotic reads identified by eggnog-mapper. We kept reads all prokaryotic reads as well as those that were not assigned to any Domain. Percentage of eukaryotic-free reads are shown in Supplementary Table S1. Median eukaryotic-free reads in metaGs are 74.5% and 15.6% in sediment/rhizosphere and root samples, respectively.

Reviewer 1, comment 27:

Line 577-580. Idba-ud is generally a poor choice for assembly of metagenomes. Did you try other metagenome assemblers, such as Spade? Co-assemblies do a very poor job of assembling true MAGs especially with closely related species/strains found in the different samples, You likely get a lot of artificial recombination events happening. I do not recommend AT ALL. How many samples/MAGs were generated with co-assemblies?

Reply: Respectfully, we do not agree with the comment made by the reviewer. Idba-ud is a widely used assembly program and very stable. Since its publication in 2012, the idba-ud paper has been cited 2793 times, and more than 200 times in 2023 alone. Furthermore, we assembled a subset of our metagenomes using spades to demonstrate that the assembly algorithm did not change the identity and quality of our assembled MAGs. On the contrary, we got better completeness in MAGs assembled by idba-ud when compared to metaspades (for the same assembled genomes). ANI values between same genomes assembled independently by spades and idba-ud were very high: Median: 99.86% Min: 99.44% Max: 100%. This indicates that both assemblers were producing almost identical contigs within MAGs.

Fig.1 Quality score, completeness, contamination and GC content of MAGs independently assembled by spades or idba_ud for a subset of our metagenomes. All MAGs that were independently generated using both algorithms were included in this analysis. Quality Score and Completeness were statistically greater for MAGs binned using idba_ud contigs than meta-spades (ANOVA: $p < 0.01$).

On the idea of co-assemblies, we also argue that is a common practice in microbiome science. Co-assemblies are particularly helpful to improve the representation of MAGs in studies like ours in which high diversity prevents the assembly and binning of a large proportion of the prokaryotic diversity. For recent examples in the literature please look at Molari et al., 2023; Zhang et al., 2022; Wilkinson et al., 2020; Rodriguez-R et al., 2020; Garner et al., 2023 just to cite a few of high-profile papers utilizing co-assemblies in contrasting environments (full citations at the end of this document). We understand the concern raised by the reviewer, that performing co-assemblies could blur differences at the sub-species level across the studied gradients. However, since the scope of the present study is at the community and genomospecies level, our results are not biased by this approach. Eco-evolutionary processes at the population/sub-species level, which may be biased by co-assemblies, are beyond the scope of this study. Further, we have assembled MAGs from both co-assemblies and individual assemblies that are part of the same genomospecies, including *Ca. Thiodiazotropha gsp 31*, *gsp 32*, and *gsp 33*. The ANI between co-assembled and individually assembled MAGs for these three cases are 97.09%, 99.23%, and 99.94%, respectively. The ability to recover the same prokaryotic species by the two

approaches suggest that co-assemblies can be used to successfully recover genomes from our metagenomes. All MAGs are a consensus representation of a prokaryotic species within a complex metagenomic sample. In a co-assembly, the only difference is that the assembly is performed from a population at a larger spatial scale. Thus, we argue that co-assembled MAGs are as real/artificial as a MAGs from an individual metagenome.

Reviewer 1, comment 28:

Line 600. There are other estimates of genome coverage in metagenomes; have you tried CoverM?

Reply: We calculated coverage using CoverM and got very similar results to our published method ($R^2 = 0.980$, $p < 0.001$, see Rodriguez-R et al. 2020). Please see figure below:

Figure 2: Relation between relative abundance of MAGs as calculated using the default parameters in coverM and published method by Rodriguez-R et al. (2020) for all our MAGs in all assessed samples.

Reviewer 1, comment 29:

Line 620-621. % of reads kept/used/identified per metagenome?

Reply: Similar to what we have reported before, the roots from the short phenotype had a greater percentage of prokaryotic reads mapped by megablast with our

stringent conditions (median: 17.4%), whereas the tall roots only mapped 5.4%. This was added between lines 1021-1022. We acknowledge that especially for the tall phenotype this is a low percentage. However, the conclusions of our manuscript are not affected by this issue. For instance, short reads analysis also points to sulfur oxidation being a key process coupled to nitrogen fixation in the short *Spartina* phenotype (Gammaproteobacteria and Chromatiales g spp contributing more *nif* transcripts in the short phenotype compared to the tall, see Supplementary Figure S7). Furthermore, RNA amplicon analysis from the same samples as the ones in which we did metaT, also showed a greater *Ca. Thiodiazotropha* relative transcript abundance in the short vs tall roots (see below). This figure has been added to supplementary materials and cited in the manuscript within lines 363-366 (Supplementary Figure S10).

Reviewer 1, comment 30:

Line 625-628. Why not map to the reference genome and just use DESeq2 or edgeR or other method to determine differential gene expression?

Reply: A normalization approach is needed before doing any statistic due to differences in sequencing depth and gene size across metatranscriptomic samples. We normalized based on the methods published in Salazar et al., 2019. DESeq2 or edgeR could be used but there are problematic for non-isolate RNA-seq data (personal communication with the author of DESeq2). Thus, we preferably decided to use the Mann-Whitney test, a robust non-parametric test, when comparing different conditions in the marsh.

Reviewer 1, comment 31:

Line 668-669. But mixing primer sets at that level might lead to erroneous conclusions - some regions might resolve at genus or species level, others family or order levels.

Reply: It is true that different primer sets have biases against different taxa at different biological units. Thus, it is even more impressive that a coherent grouping from marine environments emerged in beta-diversity while including that background noise. For instance, a seagrass amplicon dataset used the 341F/806R primer set, while a mangrove dataset used the 338F/806R. All other marine studies used the 515F/806R primer set. And even then, all samples from seagrass and coastal wetlands clustered together -Figure 6a. This indicates that the difference in the root microbiomes of terrestrial and marine samples is so profound that even when using different primer sets, a clear separation is observed. The majority of the studies -from both terrestrial and marine sites- used the 515F/806R primer dataset (55%). Furthermore, we selected studies that used primer sets only from the V3-V4 region (338F/806R, 341F/806R, 515F/806R, 520F/799R, 341F/785R). This information was added to the manuscript between lines 1100-1101.

Reviewer 1, comment 32:

Line 675-677. Please provide a bit more detail here.

Reply: As mentioned in a response before, putative function was based on taxonomy at the genus level. We curated a list of genera with known sulfur/sulfate reducing and sulfur oxidizing metabolism. The list was included as supplementary information in Rolando et al., 2022. However, for easier access, we have also included the list and references to the supplementary tables of this manuscript. Please see Supplementary Table S12.

Reviewer #2 (Remarks to the Author):

Reviewer 2, comment 1:

The authors have performed a very interesting study, executed in a solid way and written nicely and clear. The study builds well on previous work done by the authors providing novel insight in how sulfide stress is handled by a foundation plant through the interaction

with sulfur cycling bacteria that also seem to have the additional benefit for the host that they fix nitrogen. The study is descriptive but uses contrasting conditions in a field situation to demonstrate the interaction. Experimental proof could be obtained by reciprocal transplantation experiments, seedling work as well as by manipulation of sulfide conditions. The work is novel, the host-sulfur cycling microbes is mainly known from the deepsea and animal hosts. This study together with some previous work on seagrasses strongly suggest that they also play an important role in plants living at the interface with the sea. The combination of approaches combined in this work is impressive with sufficient but limited replication.

Reply: We appreciate the positive and constructive comments of the reviewer. We agree that future studies on this system could include more experimental work such as a reciprocal transplant experiment, and greenhouse experiments with manipulative conditions. Please note that all referenced lines in the reply letter are linked to the track-changes PDF document.

Reviewer 2, comment 2:

The work supports the conclusions and claims as presented. The used methodology is sound, though some chosen thresholds are arguable. For example, the MAGs used in this study are considered high quality at a completeness of 50% while generally this is so for 90%. It is unclear to what extent the story remains standing if the general threshold is used. I think the authors need to demonstrate the (lack of) effect of this threshold for the story of the paper.

Reply: We kept MAGs with quality score (QS) greater than > 50 as is common in several microbiome papers, see for example Parks et al., 2017. QS is defined as $\text{Completeness} - 5 \times \text{Contamination}$ (please see again Parks et al., 2017). Bowers et al., (2017) classify MAGs up to Medium-Quality when they have more than 50% completeness and less than 10% contamination (i.e., $\text{QS} = 0$). Our study has more stringent requirements than Bowers et al., 2017, since we require a $\text{QS} > 50$. However, because Bowers et al., also require the presence of rRNA genes and tRNA genes for classifying a MAG as HQ, we decided to remove that term "high-quality" from our manuscript. Out of the 239 MAGs, 58 were had greater than 90% completeness and less than 5% contamination such as MAGs relevant for the story of the manuscript, including the highly abundant and active *Ca. Thiodiazotropha* gsp. 31 and gsp. 33, and sulfate reducer gsp 134. On the other hand, *Desulfosarcinaceae* gsp. 68 and gsp. 80 have a completeness of 84.0% and 64.2%, and contamination of 6.6% and 1.9%, respectively. Contamination, completeness and quality scores for all MAGs can be found in Supplementary Table S2.

Reviewer 2, comment 3:

In the performed Mann-Whitney test p-values < 0.5 are considered significant, is there a typo here or is there a sound argument to use this threshold?

Reply: Sorry, yes, this was a typo. The p-value was < 0.05 for all cases. Thanks for noticing. This was changed in all pertinent figures.

Reviewer 2, comment 4:

The method section is clear and provides enough details so that the work can be reproduced.

Reply: Thanks for the positive comment.

Reviewer 2, comment 5:

For detailed feedback and small comments and remarks, please see the attached pdf.

It seems highly likely that the process identified here is an important driver in the zonation of marine macrophytes which form very important ecosystems across the globe. This paper will certainly (re)activate the scientific community active in these ecosystem and hopefully will unite research across kingdoms to address the evolution of these interactions.

kind regards
Aschwin Engelen

Reply: Thank you again for the positive review. We have included the detailed feedbacks from the attached PDF into the manuscript. Please see replies below.

Reviewer 2, comment 6:

Title: replace by species name?

Reply: Thanks for the suggestion, title was modified to include the plant species name.

Reviewer 2, comment 7:

Line 38: where is the cycling speed measured?

Reply: We completely rewrote the abstract to reduce word counts as required by the journal editorial policy. We removed that sentence.

Reviewer 2, comment 8:

But see Cucio et al 2018 <https://doi.org/10.3389/fmars.2018.00171> for a shotgun metagenomic approach

Reply: Thanks for suggesting the 2018 paper by Cúcio et al. We have modified the introduction based on their findings as well. See for instance lines 132-134, 149-151, 705-707.

Reviewer 2, comment 9:

Line 115-116: reference required as size is not necessarily reflecting productivity

Reply: Additional references were added from previous work quantifying the physiological stress of the short *Spartina* phenotype (please see lines 196-200).

Reviewer 2, comment 10:

Line 122-123: It is a bit weird that the stress gradient theory does not comeback here

Reply: Due to comments made by reviewers 1 and 3 we decided to downplay the relevance of the stress gradient hypothesis in the manuscript. We now focus only on two contrasting conditions along the marsh.

Reviewer 2, comment 11:

Lines 137-138: is this really high quality?

Reply: High-quality was removed. However, we still kept MAGs with a quality score greater than 50, as they still provide insightful information on the studied system.

Reviewer 2, comment 12:

Line 198: how about limited sequence depth for a highly complex microbial community

Reply: Yes, we agree. These samples are highly complex, having nonpareil diversity values as high as terrestrial soils. The text was modified between lines 267-268.

Reviewer 2, comment 13:

Lines 235-236: Are there sequence differences between the genes of gsp 68 and 134?

Reply: Yes, percent identity of *dsrA*, *dsrB* and *aprA* between the two genomospecies were 82.1%, 83.2%, and 87.0%, respectively.

Reviewer 2, comment 14:

Lines 239-240: clear difference between sediment and root, but position of rhizosphere not so clear along a gradient, mostly in metaG and fig5c

Reply: Yes, indeed. We changed the phrasing to say that the diversity was lower in the root compared to both sediment and rhizosphere (please see line 240-241).

Reviewer 2, comment 15:

Line 243: unclear where this is shown in fig5

Reply: This is evidenced in the middle panel of figure 2a (former figure 5a), where a steep decrease in prokaryotic 16S rRNA gene copies is found between the rhizosphere and root compartments in the tall phenotype, but not in the short phenotype. We added the reference to panel a to make it clearer (please see new text in lines 243-246).

Reviewer 2, comment 16:

Line 250: None of them contain the permanova results

Reply: PERMANOVA results can be found in Supplementary Table S2.

Reviewer 2, comment 17:

Line 253: statistical support is lacking in this entire section

Line 254: not very convincing in fig5

Lines 258-259: Fig5c is really not suited to see these differences well as they are depicted in different plots

Reply: Yes, that is correct. We decided to omit symbols in old Figure 5 panel c because it would be too crowded if added to each one of the boxplots in this plot (280 boxplots). In order to streamline the manuscript, as suggested by Reviewer 3, we have decided to remove this section of the manuscript as it was not central to the main story of our study. We decided to move panel 5c to the supplementary materials (Now Supplementary Figure S1).

Reviewer 2, comment 18:

Line 263: specifically nifK only

Reply: Indeed, Supp. Figure S7 only shows nifK in order to avoid adding more figures to an already too crowded manuscript. But similar results, showing large transcripts of cyanobacteria in the short sediment was found also in nifH and nifD. Please see below:

Reviewer 2, comment 19:

Line: 274: pvalue of 0.5 can not be considered significant, zero missing?

Reply: Yes, indeed a zero was missing. Thank you for noticing. This was corrected in all legends.

Reviewer 2, comment 20:

Line 288: best possible term?

Reply: Sentence was rephrased. Please see new line 542-545.

Reviewer 2, comment 21:

Line 296: reference?

Reply: Thanks, references were added. Please see lines 549-552.

Reviewer 2, comment 22:

Line 310: not obvious from the plot as the reader can not see what coastal plants are represented, perhaps use symbols?

Reply: I believe the confusion was with the terminology used in previous Figure 6. Seabed was used for seagrass meadows and coastal wetlands for salt marshes and mangrove ecosystems. This was corrected by changing the name of the ecosystem type from seabed to seagrass meadow. It is not possible to add symbols for plant species due to the large number of symbols that would be needed (56 species). Please see new Figure 6.

Reviewer 2, comment 23:

Lines 336-339: see the previously mention Cucio paper

Reply: Sentenced was rephrased. Please see lines 640-642.

Reviewer 2, comment 24:

Line 372-373: Cucio et al posed the theory that lucinids can actually obtain their symbionts from seagrass

Reply: Thanks for sharing the article. Yes, we have added this into the discussion between lines 705-707.

Reviewer 2, comment 25:

Line 655: marine term more used at depth: coastal marine?

Reply: Yes, indeed. We have renamed seabed to seagrass meadows as explained in comment #22.

Reviewer 2, comment 26:

Line 666: please provide an indication of contamination

Reply: It is highly variable due to the fact that some studies did not block plant mitochondrial or plastid 16S rRNA gene amplification during PCR. That is why we removed samples with less than 5000 reads -to avoid samples with large host contamination. After all quality filtering steps, the mean number of reads left were 13,157 from the initial 20,000 subsampled reads (on average 65.8% reads kept). This was added to the methods between lines 1096-1097.

Reviewer 2, comment 27:

Figure 1: not very obvious in the inset what is inside and outside the root

Reply: In Figure 1 we wanted to show that microbes closely associated to the root perform sulfur oxidation. It is currently unknown if sulfur bacteria are mainly located in the rhizoplane (root surface), or root endosphere (inside the root). Thus, it was not objective of the figure to show where exactly in the root tissue this process is happening.

Reviewer 2, comment 28:

Figure 2: should be 90% completeness and less than 5% contamination

Reply: We removed the term high-quality. Most of the key microbes described in this study had greater quality than 90% completeness and 5% contamination anyway. Including Thiodiazotropha gsp31, gsp33, and Desulfosarcinaceae gsp134.

Reviewer 2, comment 29:

Figure 3: difference of symbols between short and tall is not obvious enough

Reply: Symbols were changed to make all figures in the manuscript more readable. Please see new Figure 4 (old Figure 3).

Reviewer 2, comment 30:

Supplementary Figure S8: number of replicates is unclear here, plus output of the test assumption test (data is non-normal)

Reply: Number of replicates was added to the figure legend (n = 4 per compartment and Spartina phenotype). Mann-Whitney tests are indeed usually used in non-normal data. However, the test itself does not have that assumption. Non-parametric tests

can be also applied to normally distributed datasets. Please see Supp. Figure S11 (Old Figure S8)

Reviewer 2, comment 31:

Supplementary Figure S10: species name in italics

Reply: Thanks for noticing. Old Supp. Figure S10 was removed to streamline the manuscript as suggested by Reviewer 3.

Reviewer #3 (Remarks to the Author):

Reviewer 3, comment 1:

The submitted manuscript " Sulfur oxidation and reduction are coupled to nitrogen fixation in the roots of a salt marsh foundation plant species " explores the coupling of S cycling and N fixation during coastal wetland plant-microbe interactions, using tools from molecular microbial ecology and biogeochemistry. The study is ecologically intriguing, but before I can recommend its publication, I believe several improvements are necessary:

Reply: We appreciate the thorough review and believe that we have improved the manuscript by addressing nearly all of the reviewer's comment. Please note that all referenced lines in the reply letter are linked to the track-changes PDF document.

Reviewer 3, comment 2:

1. Streamlining and focusing the manuscript: The authors should seriously streamline the manuscript by focusing on the key story they want to convey. Currently, there are multiple stories being told simultaneously. For instance, in the introduction, the authors delve into the stress gradient hypothesis, which might not be central to their study as no gradients were sampled, only two extremes along a hypothesized gradient. Additionally, aspects on C fixation are included in the introduction and discussion, which seem highly speculative based on the data. The meta-analysis comparing microbial communities across ecosystem types, while interesting, appears unnecessary for delivering the main story and could be used to prepare a separate manuscript. The main manuscript includes 6 figures, and there are another 10 figures in the supplement, many of which have more than 3 panels, further contributing to the manuscript's complexity.

Reply: We appreciate the reviewer's comment. We have removed reference to the stress gradient hypothesis, as it also was brought up by reviewer 1. We have reduced the intro and discussion on C fixation, as indeed this manuscript mainly focuses on the coupling of N fixation to the S cycle. Similarly, we removed the section on potential nitrification rates as this was not central to the main story of the manuscript. However, we argue that Figure 6 is needed in order to show that the root

microbiomes of salt marsh plants, such as *Spartina alterniflora* are highly similar to those found in seagrass and mangrove ecosystems, highlighting that this described process most likely has global relevance. The meta-analysis presented in Figure 6 also addresses the need for broadening the scope of the manuscript, as requested by Reviewer 1.

Reviewer 3, comment 3:

2. Addressing novelty of findings: The manuscript's novelty is not clearly demonstrated due to the many aspects touched upon. The authors need to address specific concerns about the novelty of their findings. You state that previous studies have shown that *Ca. Thiodiazotropha* bacteria are not just found in lucinid clams but also inhabit the roots of a diverse array of seagrass species along with that of the coastal cordgrass *S. alterniflora* (Martin et al., 2020a; Rolando et al., 2022). Also, your statement in L317-321 reads like other studies already captured the key findings of your study: "Similar results have been obtained (..) in seagrass and other coastal wetland plant species, where nitrogen fixation genes and their expression have been affiliated with microorganism closely related to sulfate-reducing and sulfur oxidizing bacteria (Thomas et al., 2014; Crump, et al. 2018; Kolton et al., 2020)". The authors should clarify the ecological novelty of their results and emphasize how they contribute to the existing body of knowledge.

Reply: We agree that the novel aspects of our study could have been more clearly stated in the submitted version of our manuscript. It is correct that this manuscript builds upon previously published data. However, this is the first manuscript presenting genomic and gene expression evidence for the coupling of sulfur oxidation to nitrogen fixation in the roots of a marine plant -and by extension, any plant species to the best of our knowledge. Indeed, few genome-centric studies have been performed and microbial communities have rarely been partitioned by plant compartment in coastal marshes. Of the few previous studies, including ours (Kolton et al., 2020 and Rolando et al., 2022) conclusions were based on amplicon analysis or only using metatranscriptomic short reads, without having gene expression profiles of reconstructed genomes, as performed in this study. To the best of our knowledge, this is the first study providing genomes from diazotrophic sulfur-oxidizing bacteria retrieved from the root tissue of any plant, plus paired with gene expression. This allowed us to elucidate the activity of microbial groups at the species level (within the same genomes) that mediate the coupling of sulfur oxidation (and sulfate reduction) to nitrogen fixation. We also used contrasting growing conditions to show how under greater sulfidic stress, the strength of the symbiosis between sulfur-oxidizing diazotrophs and *Spartina alterniflora* is stronger. Furthermore, the 16S rRNA gene meta-analysis provides evidence on the relevance of coupled sulfur oxidation and diazotrophy in coastal marine plants worldwide (from seagrass, marsh and mangrove ecosystems). The knowledge gaps addressed by our study and novelty are highlighted

in the introduction and discussion sections, for example between lines 134-137, 152-155, 208-216, 585-588, 590-594, 640-642, 652-662, 663-664.

Reviewer 3, comment 4:

3. Providing evidence for stressed phenotype: Throughout the manuscript, the authors refer to a stressed phenotype of *S. alterniflora*, but no evidence is provided to support this claim. Physiological measurements are needed to back this notion. Isn't it possible that the short phenotype is well-adapted to its microenvironment and not necessarily more stressed than the tall phenotype? Additionally, referring to the study as a "gradient study" might be misleading since only two positions along a transect were sampled.

Reply: We agree that we did not capture the gradient, since we sampled the extremes of the stress gradient. We do not provide direct evidence of physiological stress in this manuscript. However, there is a large amount of literature describing the stress gradient in *Spartina alterniflora* marshes. The difference in growth form of *Spartina* is attributed to a response to harsh abiotic conditions. We have added sentences from previous research providing evidence of physiological impairment in the short phenotype. In particular, it is known that growth and the photosynthetic apparatus is impaired. Similarly, nitrogen acquisition is limited due to a shift in plant anaerobic metabolism (alcohol fermentation). Further, previous research has well established that more reducing conditions, elevated sulfide levels, and higher salinity occur in the short *Spartina* marsh. Please see lines 168-200.

Reviewer 3, comment 5:

4. Strengthening evidence for coupling of S oxidation and N fixation: To provide stronger evidence for the coupling of sulfide oxidation and N fixation, the authors should address the following points:

- o Correlation across the entire microbial community: The manuscript currently shows the correlation between functional gene transcription for sulfur oxidation and nitrogen fixation within a single microbial species. The authors should consider exploring correlations across the entire microbial community.

Reply: We did not find a correlation between the expression of sulfur oxidation and nitrogen fixation genes at the entire microbial community level. However, our results show that the correlation in transcript abundance of the two processes is not at the ecosystem level, but at the species level, where the coupling of biogeochemical processes occur in nature. We argue that the signal is lost at the community level because there are S-oxidizing bacteria that do not perform N fixation, as well as diazotrophs that do not perform sulfur oxidation. Thus, the activity of non-diazotrophic S-oxidizers, and non-S-oxidizers diazotrophs generates noise that prevents showing a correlation between these two processes. We argue that the benefit of using a genome-centric approach, which includes coupled metagenome and

metatranscriptome studies like ours, is that it allows us to unravel otherwise non-detectable processes measured at larger ecological scales.

Reviewer 3, comment 6:

o Similar N fixation rates between phenotypes: The authors need to explain why N fixation rates do not significantly differ between *S. alterniflora* phenotypes. Additionally, Figure 2b presents root vs. sediment but not the rhizosphere (unlike panel a). The authors should clarify this discrepancy.

Reply: We did find greater N fixation rates in the root compartment of the stressed short phenotype under both oxic and anoxic conditions as shown in Figure 4a and between lines 305-307. The reviewer is probably referring to the expression level of the nifK gene, which indeed was not significantly different between the studied phenotypes in the root compartment. As mentioned in the text (lines 236-238, and 834-835), we did not have enough rhizosphere sample to perform metatranscriptomic analysis. That is why there is no gene expression analysis performed in that compartment.

Reviewer 3, comment 7:

o Lack of $\delta^{15}\text{N}$ figure: Given that the study primarily focuses on N fixation, the absence of a figure on natural abundance $\delta^{15}\text{N}$ of plant tissue is surprising. Substantial N_2 fixation by root microbes could be reflected in $\delta^{15}\text{N}$ (e.g. Högberg 1997 New Phytol). Likewise, differences in N_2 fixation between the two phenotypes should be reflected in the natural abundance of ^{15}N in plant tissues.

Reply: Thank you for this comment! We have added a figure showing our results from the natural abundance isotopic composition of nitrogen, $\delta^{15}\text{N}$, as suggested by the reviewer. The $\delta^{15}\text{N}$ of the stressed short phenotype, where we measured greater rates of N fixation, also had lower $\delta^{15}\text{N}$. This provides additional evidence of greater cumulative N fixation in the marsh dominated by short phenotype of *Spartina alterniflora*. Please see lines 307-323, and new right panel in Figure 4a.

Reviewer 3, comment 8:

o Consideration of sulfide oxidation as a measurable process: Contrary to the authors' claim, sulfide oxidation is not an "unmeasurable process" as suggested in Findlay et al. 2020 (). This means, it would have been possible to provide more biogeochemical evidence for the coupling of N fixation and sulfide oxidation.

Reply: We agree that quantifying sulfide oxidation is certainly possible but nonetheless difficult and requires specialized expertise and equipment. As per the reviewer's comment, we have deleted the lines in our manuscript that mention sulfide oxidation as an unmeasurable process. Further, we carefully read the paper provided

by the reviewer. We acknowledge that sulfide oxidation to sulfate can be measured. However, the referenced paper does not quantify other pathways of sulfur oxidation (e.g., thiosulfate and elemental sulfur oxidation). The study only quantifies sulfide oxidation to sulfate, and intermediate oxidation states of S were also not measured. The referenced study underscores the fact that sulfur oxidation remains a very cumbersome process to measure *in situ*. The study tracks the reactivity of 5 different radioactive sulfur pools. Furthermore, the study was done in intact cores, and a main objective of our study is to detect differences between belowground compartments. Separating the roots from the surrounding sediment would add an extra layer of complication for measurements like this. But we agree that direct measurements of sulfur oxidation would be ideal. Future work should definitively perform manipulative experiments, as also brought up by reviewer 2, to improve mechanistic understanding of S-oxidation in this system.

References:

1. Bowers, R.M., Kyrpides, N.C., Stepanauskas, R., Harmon-Smith, M., Doud, D., Reddy, T.B.K., Schulz, F., Jarett, J., Rivers, A.R., Eloie-Fadrosch, E.A. and Tringe, S.G., 2017. Minimum information about a single amplified genome (MISAG) and a metagenome-assembled genome (MIMAG) of bacteria and archaea. *Nature biotechnology*, 35(8), pp.725-731.
2. Cúcio, C., Overmars, L., Engelen, A.H. and Muyzer, G., 2018. Metagenomic analysis shows the presence of bacteria related to free-living forms of sulfur-oxidizing chemolithoautotrophic symbionts in the rhizosphere of the seagrass *Zostera marina*. *Frontiers in Marine Science*, 5, p.171.
3. Cúcio, C., Engelen, A.H., Costa, R. and Muyzer, G., 2016. Rhizosphere microbiomes of European seagrasses are selected by the plant, but are not species specific. *Frontiers in microbiology*, 7, p.440.
4. Garner, R.E., Kraemer, S.A., Onana, V.E., Fradette, M., Varin, M.P., Huot, Y. and Walsh, D.A., 2023. A genome catalogue of lake bacterial diversity and its drivers at continental scale. *Nature microbiology*, pp.1-15.
5. Kolton, M., Rolando, J.L. and Kostka, J.E., 2020. Elucidation of the rhizosphere microbiome linked to *Spartina alterniflora* phenotype in a salt marsh on Skidaway Island, Georgia, USA. *FEMS microbiology ecology*, 96(4), p.fiaa026.
6. Martin, B.C., Middleton, J.A., Fraser, M.W., Marshall, I.P., Scholz, V.V., Hausl, B. and Schmidt, H., 2020. Cutting out the middle clam: lucinid endosymbiotic bacteria are also associated with seagrass roots worldwide. *The ISME journal*, 14(11), pp.2901-2905.
7. Mendelssohn, I.A., Morris, J.T., 2000. Eco-physiological controls on the productivity of *Spartina alterniflora* Loisel. *Concepts and controversies in tidal marsh ecology*, pp.59-80.

8. Molari, M., Hassenrueck, C., Laso-Pérez, R., Wegener, G., Offre, P., Scilipoti, S. and Boetius, A., 2023. A hydrogenotrophic *Sulfurimonas* is globally abundant in deep-sea oxygen-saturated hydrothermal plumes. *Nature Microbiology*, 8(4), pp.651-665.
9. Parks, D.H., Rinke, C., Chuvochina, M., Chaumeil, P.A., Woodcroft, B.J., Evans, P.N., Hugenholtz, P. and Tyson, G.W., 2017. Recovery of nearly 8,000 metagenome-assembled genomes substantially expands the tree of life. *Nature microbiology*, 2(11), pp.1533-1542.
10. Rodriguez-R, L.M., Tsementzi, D., Luo, C. and Konstantinidis, K.T., 2020. Iterative subtractive binning of freshwater chronoserries metagenomes identifies over 400 novel species and their ecologic preferences. *Environmental Microbiology*, 22(8), pp.3394-3412.
11. Rolando, J.L., Kolton, M., Song, T. and Kostka, J.E., 2022. The core root microbiome of *Spartina alterniflora* is predominated by sulfur-oxidizing and sulfate-reducing bacteria in Georgia salt marshes, USA. *Microbiome*, 10(1), p.37.
12. Salazar, G., Paoli, L., Alberti, A., Huerta-Cepas, J., Ruscheweyh, H.J., Cuenca, M., Field, C.M., Coelho, L.P., Cruaud, C., Engelen, S. and Gregory, A.C., 2019. Gene expression changes and community turnover differentially shape the global ocean metatranscriptome. *Cell*, 179(5), pp.1068-1083.
13. Wilkinson, T., Korir, D., Ogugo, M., Stewart, R.D., Watson, M., Paxton, E., Goopy, J. and Robert, C., 2020. 1200 high-quality metagenome-assembled genomes from the rumen of African cattle and their relevance in the context of sub-optimal feeding. *Genome biology*, 21, pp.1-25.
14. Zhang, L., Jonscher, K.R., Zhang, Z., Xiong, Y., Mueller, R.S., Friedman, J.E. and Pan, C., 2022. Islet autoantibody seroconversion in type-1 diabetes is associated with metagenome-assembled genomes in infant gut microbiomes. *Nature communications*, 13(1), p.3551.

Reviewer #1 (Remarks to the Author):

This manuscript is important in relating the link between chemoautotrophic processes involving sulfur oxidation and nitrogen fixation. It also underscores the relationship between sulfur oxidizing and sulfur reducing microbes in the rhizobiome of coastal seagrasses. This revision is much improved from the last one. However, I still have a few comments that I believe would improve the manuscript.

Lines 104-106. The authors stated that they removed the stress hypothesis from the revised manuscript. However, the statements here do not reflect that. Are there more beneficial microbes associated with the root zone in the short vs. tall *S. alterniflora* plants? Are there more microbes in general in those roots? It is likely that there are different microbes, depending on what the plant and microbes need, and there are likely just as many 'beneficial' microbes in both instances. Maybe there are more sulfur oxidizers and different sulfate reducers, but less of other 'beneficial' microorganisms that are found in the other plants.

Line 201-202. I understand if this didn't get measured, but it would have been great to concurrently look at the transcription of sulfur oxidizing and sulfur reducing genes under aerobic and anaerobic conditions, along with rates of CO₂ fixation.

Line 207. I noticed earlier you used the term 'transcriptomic' and here is it 'metatranscriptomic' Please be consistent.

Line 219-221. But the MAGs are only ~10-20% of prokaryotic reads. Why not see what the fraction of reads map to *nifK* and *dsrA* assembled/annotated genes in the metagenome compared to the metatranscriptome? Maybe compare to a constitutively expressed gene?

Line 225. Please remind the reader that these are the sulfur-oxidizer and sulfur-reducer MAGs.

Line 230. This analysis would be infinitely better with assembled but unbinned reads instead of short reads. That way, you could infer and resolve the taxonomic affiliations much better.

Line 257-262. It is hard to say that they don't all have all these genes, since their genomes are not complete, and it is my understanding that most are not high quality either.

Line 329. Again, these MAGs represent only a small % of the microbes present.

Line 366-367. They are not novel to this study, where they have been found in the root zone before.

Reviewer #3 (Remarks to the Author):

All my comments have been adequately addressed by the authors. A very nice study!

Peter Mueller

REVIEWER COMMENTS

Reviewer #1 (Remarks to the Author):

Reviewer 1, comment 1:

This manuscript is important in relating the link between chemoautotrophic processes involving sulfur oxidation and nitrogen fixation. It also underscores the relationship between sulfur oxidizing and sulfur reducing microbes in the rhizobiome of coastal seagrasses. This revision is much improved from the last one. However, I still have a few comments that I believe would improve the manuscript.

Reply: We are glad that the reviewer identified improvements in our manuscript following the first round of peer review. We also value the additional comments provided here. Please find a detailed response in this reply letter and be aware that the lines mentioned in this reply letter correspond to the track-changes document.

Reviewer 1, comment 2:

Lines 104-106. The authors stated that they removed the stress hypothesis from the revised manuscript. However, the statements here do not reflect that. Are there more beneficial microbes associated with the root zone in the short vs. tall *S. alterniflora* plants? Are there more microbes in general in those roots? It is likely that there are different microbes, depending on what the plant and microbes need, and there are likely just as many 'beneficial' microbes in both instances. Maybe there are more sulfur oxidizers and different sulfate reducers, but less of other 'beneficial' microorganisms that are found in the other plants.

Reply: We acknowledge the comment made by the reviewer. Similar to what the reviewer is stating, we hypothesize that beneficial microorganisms inhabiting the root microbiome of the short phenotype may be relevant to alleviate environmental stress. We modified the text to specifically talk about plant-microbe interactions related to amelioration of stress. In particular, in our study, we assess the oxidation of sulfide, a well-described phytotoxin. Please see lines 105-107. The broader question if there is more or less positive-positive interactions as a function of environmental stressed is beyond the scope of this manuscript, which mainly focuses of the cycling of S and N. We did not measure other important plant microbiome functions such as pathogen suppression, synthesis of plant hormones, or acquisition of other nutrients besides N to fully address this question.

Reviewer 1, comment 3:

Line 201-202. I understand if this didn't get measured, but it would have been great to concurrently look at the transcription of sulfur oxidizing and sulfur reducing genes under aerobic and anaerobic conditions, along with rates of CO₂ fixation.

Reply: We agree. Unfortunately, this was not measured in this study, as we did not have enough material for metatranscriptomic analysis after incubations. We will definitively look closer into chemoautotrophic C fixation in the roots of coastal marine plants in future work.

Reviewer 1, comment 4:

Line 207. I noticed earlier you used the term 'transcriptomic' and here is it 'metatranscriptomic' Please be consistent.

Reply: Thank you for catching this up. Metatranscriptomic is now used all along the manuscript. Please see line 145.

Reviewer 1, comment 5:

Line 219-221. But the MAGs are only ~10-20% of prokaryotic reads. Why not see what the fraction of reads map to nifK and dsrA assembled/annotated genes in the metagenome compared to the metatranscriptome? Maybe compare to a constitutively expressed gene?

Reply: To improve the representation of this analysis, and as suggested by the reviewer in comment 7, we decided to include unbinned scaffolds into this analysis.

The extra analysis was performed as following:

- 1. ORFs were predicted from scaffolds from all assemblies and co-assemblies using prodigal.**
- 2. Functional annotation of predicted ORFs was performed with eggno-mapper**
- 3. We retrieved all annotated nifK and dsrA ORFs/alleles.**
- 4. We used CD-HIT-EST to cluster ORFs by 95% nucleotide identity (to avoid mapping two times to the same ORF from different assemblies).**
- 5. Alleles from MAGs were identified within the CD-HIT-EST clusters.**
- 6. We mapped metatranscriptomic short reads to CD-HIT-EST representatives with megablast blastn (percent identity > 95% and alignment > 90%).**
- 7. For normalization, we calculated the percentage of short reads annotated as nifK, and oxidative/reductive dsrA that mapped to the CD-HIT-EST cluster representatives.**
- 8. Taxonomic annotation of ORFs that were not part of MAGs was performed using kaiju with the ncbi-nr database. ORFs not classified by kaiju, were classified by eggno-mapper.**
- 9. Figure 4C was updated based on this analysis to include unbinned scaffolds.**

By performing this analysis, we were able to map on average 23.7%, 41.1%, and 39.2% of all nifK, oxidative dsrA, and reductive dsrA short reads, respectively. We acknowledge, that there is still a large proportion of the community's

metatranscriptome that is not represented by this analysis. Thus, we decided to keep the short read analysis which is not biased by issues related to low assembly (Figure 4B, Supp. Figures S7, S8 and S9). Short read analysis is coherent with the story of the manuscript -greater nitrogen fixation and sulfur oxidation associated with Gammaproteobacteria in the stressed short phenotype, whereas Deltaproteobacteria is mostly responsible for N fixation transcripts in the tall phenotype.

As suggested by the reviewer, we also performed this analysis normalizing by the median of 10 universal single-copy phylogenetic marker genes (K06942, K01889, K01887, K01875, K01883, K01869, K01873, K01409, K03106, and K03110), finding the same trend, with MAGs and scaffolds from *Candidatus* Thiodiazotropha highly expressing genes for sulfur oxidation and nitrogen fixation mainly in the roots of the short phenotype of *Spartina alterniflora*. Please see below: Thus, we decided to keep the percentage analysis for figure 4C.

We did not normalize the metatranscriptome using the metagenome short reads since samples were taken in different years, which hinders the comparison. Furthermore, we were interested in assessing the gene expression of nitrogen fixation and sulfur oxidation/reduction genes regardless of the abundance of the MAGs. In other words, our objective was to identify which members of the prokaryotic community contributed the most expression of these genes regardless of their abundance.

Please see changes in the manuscript within lines 220-227, 230-232, 640-651 for results and methods from this extra analysis.

Reviewer 1, comment 6:

Line 225. Please remind the reader that these are the sulfur-oxidizer and sulfur-reducer MAGs.

Reply: We appreciate the suggestion. We added a reminder of the definition of genomospecies in line 225 for the reader to recognize that we are talking about MAGs.

Reviewer 1, comment 7:

Line 230. This analysis would be infinitely better with assembled but unbinned reads instead of short reads. That way, you could infer and resolve the taxonomic affiliations much better.

Reply: We addressed this suggestion in the response to comment 5. Please see above.

Reviewer 1, comment 8:

Line 257-262. It is hard to say that they don't all have all these genes, since their genomes are not complete, and it is my understanding that most are not high quality either.

Reply: We agree with the reviewer, and a cautionary statement was added to the manuscript in lines 267-268.

Reviewer 1, comment 9:

Line 329. Again, these MAGs represent only a small % of the microbes present.

Reply: It is true that our MAGs represent a small percentage of the whole community. However, for the same reason, we have binned microbes with high abundance in our studied system, since the rare microbiome would have too little coverage for assembly. Furthermore, when mapping back metatranscriptomic short reads to functional genes such as the nitrogenase and dissimilatory sulfite reductase, up to 20 to 30% of the functional genes were mapping to our sulfur-oxidizing *Ca. Thiodiazotropha* MAGs, and sulfate-reducers from the *Desulfosarcinaceae* family. Thus, we still argue that these are one of the most highly abundant and active species in our studied system.

Reviewer 1, comment 10:

Line 366-367. They are not novel to this study, where they have been found in the root zone before.

Reply: We argue that these are novel symbionts, as they are different species from those previously described in lucinid clams (ANI < 95%). Anyway, we removed the 'novel' adjective since it could lead to confusion. Please see lines 372-373.

Reviewer #3 (Remarks to the Author):

Reviewer 3, comment 1:

All my comments have been adequately addressed by the authors. A very nice study!

Peter Mueller

Reply: We are pleased that the reviewer was satisfied with our previous response. We value the feedback and positive evaluation of our work.

Reviewer #1 (Remarks to the Author):

I have read through the revised manuscript and am generally good with the revisions. However, I note two small things that should be modified in the manuscript:

Figure 2b. What is the difference between the open and filled circles? I assume it is the short vs. tall phenotype, but that designation is not in the legend at the top of the figure or in the figure caption.

Line 241. Highest transcript abundance compared to what?

Otherwise, I appreciate the authors taking my comments into account.

Reply to reviewer comments:

REVIEWERS' COMMENTS

Reviewer #1 (Remarks to the Author):

Reviewer 1, Comment 1:

I have read through the revised manuscript and am generally good with the revisions. However, I note two small things that should be modified in the manuscript:

Reply: We are pleased to find out that we have addressed all of the previous reviewer's comments.

Reviewer 1, Comment 2:

Figure 2b. What is the difference between the open and filled circles? I assume it is the short vs. tall phenotype, but that designation is not in the legend at the top of the figure or in the figure caption.

Reply: Yes, indeed it is the tall vs short phenotype. We modified the legend figure to include this. Please see new Figure 2.

Reviewer 1, Comment 3:

Line 241. Highest transcript abundance compared to what?

Reply: To all other assessed compartments (i.e., the roots of the tall phenotype and the two sediment groups). We modified the text to include this. Please see lines 240-244.

Reviewer 1, Comment 4:

Otherwise, I appreciate the authors taking my comments into account.

Reply: You're welcome. We appreciate your helpful comments.